# Analyzing and Improving Generative Adversarial Training for Generative Modeling and Out-of-Distribution Detection

## Abstract

Generative adversarial training (GAT) is a recently introduced adversarial defense method. Previous works have focused on empirical evaluations of its application to training robust predictive models. In this paper we focus on theoretical understanding of the GAT method and extending its application to generative modeling and out-of-distribution detection. We analyze the optimal solutions of the maximin formulation employed by the GAT objective, and make a comparative analysis of the minimax formulation employed by GANs. We use theoretical analysis and 2D simulations to understand the convergence property of the training algorithm. Based on these results, we develop an unconstrained GAT algorithm, and conduct comprehensive evaluations of the algorithm's application to image generation and adversarial out-of-distribution detection. Our results suggest that generative adversarial training is a promising new direction for the above applications.

## 1 Introduction

Generative adversarial training (GAT) (Yin et al., 2020) is a recently introduced defense mechanism that could be used for adversarial example detection and robust classification. The defense consists of a committee of detectors (binary discriminators), with each one trained to discriminate natural data of a particular class from adversarial examples perturbed from data of other classes. Like most other work in the area of robust machine learning, the defense is specially designed for defending against norm-constrained adversaries — adversaries that are constrained to perturb the data up to a certain amount as measured by some norm. The defense's robustness is achieved by training each detector model against adversarial examples produced by the norm-constrained PGD attack (Madry et al., 2017).

**Existing work: training and evaluating robust predictive models** A detector trained with GAT has strong interpretability — an unbounded attack that maximizes the detector's output results in images that resemble the target class data — this suggests the detector has learned the target class data distribution. However, all previous works (Yin et al., 2020; Tramer et al., 2020) focus on the empirical evaluations of GAT's application to training robust predictive models; *a theoretical understanding of why this training method causes the detector to learn the data distribution is missing.*

**This work: theoretical understanding, improved training algorithm, and extended applications** In order to better understand the GAT method, we first analyze the optimal solutions of the training objective. We start with a *maximin* formulation (eq. (5)) of the objective, and try to connect it with the *minimax* formulation (eq. (1)) that is employed by GANs (Goodfellow et al., 2014). We find that the differences between solutions of these two formulations become immediately clear when we take a game-theory perspective. We then use theoretical analysis and 2D simulations to understand the convergence property of the GAT training algorithm. Building upon these theoretical and experimental insights, we develop an unconstrained GAT algorithm, and apply it to the tasks of generative modeling and out-of-distribution detection. We find the maximin-based generative model to be more stable to train than its minimax counterpart (GANs), and at the same time more flexible as it does not have a fixed generator and can transform arbitrary inputs to the target distribution data, which might be particularly useful for certain applications (e.g., face manipulation). The model

trained with the unconstrained GAT algorithm also outperforms several state-of-the-art methods on the task of adversarial out-of-distribution detection. In summary, our key contributions are:

- We analyze the optimal solutions of the GAT objective and convergence property of the training algorithm. We discuss the implications of these results on improved training of robust predictive models, generative modeling, and out-of-distribution detection.

- We develop an unconstrained generative adversarial training algorithm. We conduct a comprehensive evaluation of the algorithm's application to image generation and adversarial out-of-distribution detection.

- Our comparative analysis of the maximin and minimax problem clarifies misconceptions and provides new insights into how they could be utilized to solve different problems.

## 2 RELATED WORK AND BACKGROUND

**Generative adversarial networks (GANs)**   The GANs framework (Goodfellow et al., 2014) learns a generator function $G$ and a discriminator function $D$ by solving the following *minimax* problem

$$\min_G \max_D V(D,G) = \mathbb{E}_{\mathbf{x} \sim p_{\text{data}}}[\log D(x)] + \mathbb{E}_{\mathbf{z} \sim p_z}[\log(1 - D(G(z)))]. \tag{1}$$

The generator $G$ implicitly defines a distribution $p_g$ by mapping a prior distribution $p_z$ from a low-dimensional latent space $\mathcal{Z} \subseteq \mathbb{R}^z$ to the high-dimensional data space $\mathcal{X} \subseteq \mathbb{R}^d$. $D : \mathcal{X} \to [0, 1]$ is a function that discriminates the target data distribution $p_{\text{data}}$ from the generated distribution $p_g$. The minimax problem is solved by alternating between the optimization of $D$ and optimization of $G$; under certain conditions, the alternating training procedure converges to a solution where $p_g$ matches $p_{\text{data}}$ (Jensen-Shannon divergence is zero), and $D$ outputs $\frac{1}{2}$ on support of $p_{\text{data}}$.

**Generative adversarial training (GAT)**   The GAT method (Yin et al., 2020) is designed for training adversarial examples detection and robust classification models. In a $K$ class classification problem, the robust detection/classification system consists of $K$ base detectors, with each one trained by minimizing the following objective

$$L(D) = -\mathbb{E}_{\mathbf{x} \sim p_k}[\log D(x)] - \mathbb{E}_{\mathbf{x} \sim p_{-k}}[\log(1 - \max_{x' \in \mathbb{B}(x,\epsilon)} D(x'))]. \tag{2}$$

In the above objective, $p_k$ is $k$-th class's data distribution, $p_{-k}$ is the mixture distribution of all other classes: $p_{-k} = \frac{1}{K-1} \sum_{i=1,\dots,K,i \neq k} p_i$, and $\mathbb{B}(x, \epsilon)$ is a neighborhood of $x$: $\{x' \in \mathcal{X} : \|x' - x\|_2 \leq \epsilon\}$. The objective is characterized by an *inner maximization* problem and an *outer minimization* problem; when the inner maximization is perfectly solved and $D$ achieves a vanishing loss, $D$ becomes a perfectly robust model capable of separating data $p_k$, from *any* $\epsilon$-constrained adversarial examples perturbed from data of $p_{-k}$. A committee of $K$ detectors then provides a complete solution for detecting any adversarial example perturbed from an arbitrary class. Objective 2 is solved using a alternating gradient method (Algorithm 1), with the first step crafting adversarial examples by solving the inner maximization, and the second step improving the $D$ model on these adversarial examples.

Clearly, the detector's robustness depends on how well the inner maximization is solved. Despite the fact that $D$ is a highly non-concave function when it's parameterized by a deep neural network, Madry et al. (2017) showed that the inner problem could be reasonably solved using projected gradient descent (PGD attack) — a first-order method that employs the following iterative gradient update rule (at initialization $x^0 \leftarrow x$, we consider $L^2$-based attack)

$$x^{i+1} \leftarrow \texttt{Proj}(x^i + \gamma \frac{\nabla \log D(x^i)}{\|\nabla \log D(x^i)\|_2}), \tag{3}$$

where $\lambda$ is some step size, and $\texttt{Proj}$ is the operation of projecting onto the feasible set $\mathbb{B}(x, \epsilon)$. The *normalized steepest ascent* rule inside the $\texttt{Proj}$ function, was introduced for dealing with the issue of vanishing gradient when optimizing with the cross-entropy loss (Kolter & Madry, 2019). The PGD attack also employs random restarting to improve its effectiveness. The idea is that for a input $x$, first generate a set of randomized inputs by uniformly sampling from $\mathbb{B}(x, \epsilon)$, perform PGD attack on each of them, and use the most effective one as the actual attack.

A review of related work on out-of-distribution detection in provided in Appendix A.

---

**Algorithm 1** GAT Detector Training Method (The Maximin Problem Solver)

---

1: Sample minibatch of $m$ samples $\{x_1^k, \ldots, x_m^k\}$ from $p_k$, and $m$ samples $\{x_1^{-k}, \ldots, x_m^{-k}\}$ from $p_{-k}$.
2: Compute adversarial examples $\{x_1', \ldots, x_m'\}$ by solving $\max_{x' \in \mathbb{B}(x,\epsilon)} D(x')$ for each $x_i^{-k}$.
3: Train the detector by minimizing $\frac{1}{m} \sum_{i=1}^{m} \left[ -\log D(x_i^k) - \log(1 - D(x_i')) \right]$ (single step).
4: Return to step 1.

---

## 3 THEORETICAL RESULTS

In this section we first reformulate objective 2 into a *maximin* problem, and then analyze the optimal solutions of the maximin problem and convergence property of Algorithm 1. We then discuss the optimal solution of the corresponding *minimax* formulation and the differences between the solutions of these two formulations. The popular generative modeling approach of GANs learns a data distribution by solving the minimax problem, but there seems to be a misconception about the differences between solutions of these two problems, and as a result, a false impression that the GANs algorithm could solve the maximin problem (Goodfellow (2016), section 5.1.1). Our analysis of optimal solutions is based on a game-theory interpretation of these problems, and the differences between these solutions are immediately clear under such an analysis.

### 3.1 THE MAXIMIN PROBLEM

In this section we provide an analysis of the optimal solutions of objective 2. Maximizing $D$ is equivalent to minimizing $\log(1 - D)$, hence eq. (2) is equivalent to

$$L(D) = -\mathbb{E}_{\mathbf{x} \sim p_k}[\log D(x)] - \mathbb{E}_{\mathbf{x} \sim p_{-k}}[\min_{x' \in \mathbb{B}(x,\epsilon)} \log(1 - D(x')))]. \qquad (4)$$

For the convenience of analysis, instead of using $\epsilon$-balls imposed on individual data samples, we use the notion of a common perturbation space: The perturbation space $\mathcal{S}$ is a subspace of the data space $\mathcal{X}$, and allows mass of $p_{-k}$ to be moved to any location in $\mathcal{S}$. A new distribution $p_t$ can be obtained by transporting the mass of $p_{-k}$ to appropriate locations in $\mathcal{S}$, via a transformation function $T : \mathcal{S} \to \mathcal{S}$. Utilizing the technique of random variable transformation, we can write the density function of $p_t$ as a function of $p_{-k}$: $p_t(y) = \int_{\mathcal{S}} p_{-k}(x)\delta(y - T(x))dx$. Figure 1 left panel is a schematic illustration of this phenomenon. Let $\mathcal{M}_+^1(\mathcal{S})$ be the set of distributions attainable by applying such transformations to the support of $p_{-k}$. With the notation of perturbation space, the inner problem in eq. (4) could then be interpreted as determining the distribution in $\mathcal{M}_+^1(\mathcal{S})$ that causes the highest (expected) loss of the $D$ function. Assuming $\mathbb{B}(x, \epsilon) = \mathcal{S}$, the interplay of the $D$ model and the adversary can be formulated as a *maximin* problem:

$$\max_D \min_{p_t \in \mathcal{M}_+^1(\mathcal{S})} U(D, p_t) = \mathbb{E}_{\mathbf{x} \sim p_k}[\log D(x)] + \mathbb{E}_{\mathbf{x} \sim p_t}[\log(1 - D(x))]. \qquad (5)$$

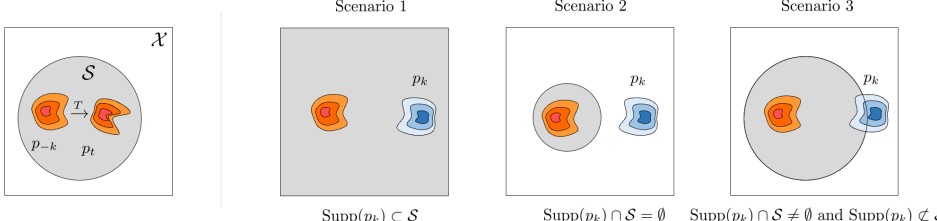

Figure 1: Left panel: a distribution $p_t$ is obtained by applying a transformation $T$ to the support of $p_{-k}$. Right panel: three scenarios to consider when analyzing problem 5. Red distribution represents $p_{-k}$ and blue distribution represents $p_k$. The data space $\mathcal{X}$ is represented by the whole space inside the square, and the perturbation space $\mathcal{S}$ is represented by the gray area.

**Optimal solutions** A convenient way of analyzing the above problem is to consider it as a two-player game: player 1 first presents different $D$ configurations, then for each $D$, player 2 determines a perturbed distribution $p_t^D$ that minimizes $U$ under the considered $D$. Then over all combinations of

Table 1: Optimal solutions for the three scenarios in Figure 1

| Scenario | Optimal solution $(D^*, p_t^*)$ |
|---|---|
| $\text{Supp}(p_k) \subset \mathcal{S}$ | $D^*$ outputs $\frac{1}{2}$ for $\text{Supp}(p_k)$ and $\leq \frac{1}{2}$ for $\mathcal{S} \setminus \text{Supp}(p_k)$; $p_t^*$ has its mass distributed to locations where $D^*$ outputs $\frac{1}{2}$. |
| $\text{Supp}(p_k) \cap \mathcal{S} = \emptyset$ | $D^*$ outputs 1 for $\text{Supp}(p_k)$ and 0 for $\mathcal{S}$; $p_t^*$ can be an arbitrary distribution in $\mathcal{M}_+^1(\mathcal{S})$. |
| $\text{Supp}(p_k) \cap \mathcal{S} \neq \emptyset$, and $\text{Supp}(p_k) \not\subset \mathcal{S}$ | For $\text{Supp}(p_k)$ outside $\mathcal{S}$, $D^*$ outputs 1, for $\text{Supp}(p_k)$ inside $\mathcal{S}$, $D^*$ outputs $\alpha = \frac{\int_{\mathcal{S}} p_k}{\int_{\mathcal{S}} p_t^* + \int_{\mathcal{S}} p_k}$ (by definition, $\int_{\mathcal{S}} p_t^* = 1$), and for other places inside $\mathcal{S}$, $D^*$ outputs $\leq \alpha$; $p_t^*$ has its mass distributed to locations where $D^*$ outputs $\alpha$. |

$(D, p_t^D)$, player 1 chooses the combination $(D, p_t^D)^*$ that gives the highest $U$ value. By analyzing both players' best strategies for playing the game, we could derive the optimal solutions (Table 1) for the three scenarios [1] depicted in Figure 1. From a game-playing perspective, the claims in Table 1 can be verified by assuming a different $D$ configuration than the claimed one, and show that there always exists a $p_t$ that results in a lower $U$ value than the $U$ value that could be achieved with the claimed $D$ configuration. Mathematical derivations of these optimal solutions are included in Appendix D.

A discussion about scenario 2 result and its implication for training robust models in provided in Appendix E.

## 3.2   THE MAXIMIN PROBLEM SOLVER

The method (Algorithm 1) for training adversarial-robust detector is in fact a solver (assuming $\mathbb{B}(x, \epsilon) = \mathcal{S}$) for the maximin problem 5: maximizing $D(x)$ is equivalent to minimizing $\log(1 - D(x))$ (step 2), and minimizing the loss is equivalent to maximizing $U$ (step 3). Algorithm 1 has the following convergence property:

**Proposition 1.** *If step 2 always perfectly solves the inner problem (i.e., the mass of $p_t$ is always moved to the location(s) where $D$ has the largest output(s)), and step 3's updates happen in $D$'s function space, and each update is sufficiently small, then the algorithm converges to the optimal solution of $D$.*

*Proof.* We consider scenario 1 in Figure 1. Let $\alpha := \max_{\mathcal{S} \setminus \text{Supp}(p_k)} D$, $A := \{x \in \mathcal{S} \setminus \text{Supp}(p_k) : D(x) = \alpha\}$, and $\beta := \max_{\text{Supp}(p_k)} D$, $B := \{x \in \text{Supp}(p_k) : D(x) = \beta\}$. We focus on the case of $1 > \alpha, \beta > \frac{1}{2}$; other cases can be proved using a similar argument. Recall that in Algorithm 1, step 2 solves the inner minimization by moving mass of $p_{-k}$ to locations where $D$ has the largest outputs, and step 3 updates $D$ by decreasing its outputs on $p_t$ and increasing its outputs on $p_k$. We further assume when mass of $p_{-k}$ is moved to multiple locations with equal $D$ outputs, the algorithm doesn't have a preference over locations (i.e., mass of $p_{-k}$ will be uniformly distributed to these locations). Algorithm 1 can be interpreted as a finite state machine that constantly switches between the following three states:

- State 1: $\alpha > \beta$. Step 2 moves the mass of $p_{-k}$ to $A$, and step 3 decreases $\alpha$ while increases $\beta$; the algorithm switches to state 2 or state 3.

- State 2: $\alpha < \beta$. Step 2 moves the mass of $p_{-k}$ to $B$, and step 3 maintains $\alpha$ while decreases $\beta$ (Appendix F.1); the algorithm switches to state 1 or state 3.

- State 3: $\alpha = \beta$. Step 2 moves the mass of $p_{-k}$ to $A \cup B$, step 3 decreases $\alpha$. Because of non-zero densities of $B$'s points on $p_k$, if $\beta$ is decreased, the decreased amount is always lower than that of $\alpha$ — the algorithm switches to state 2.

In particular, step 3 in state 1 and state 2 always results in an decrease of $\max\{\alpha, \beta\}$ (but $\beta$ cannot be decreased to below $\frac{1}{2}$, Appendix D), and step 3 in state 3 always results in an decrease of $\alpha$. The algorithm converges to the $D$ solution of $\alpha \leq \frac{1}{2}, \beta = \frac{1}{2}$. $\qquad \square$

**Practical considerations**   The above proof relies on the assumption that step 2 always *perfectly* solves the inner minimization (i.e., mass of $p_{-k}$ is always moved to the location(s) where $D$ has the

---

[1] In scenario 2 and 3, $D^*$ doesn't need to be defined on $\mathcal{X} \setminus (\mathcal{S} \cup \text{Supp}(p_k))$.

largest output(s)). In practice, as a gradient-based search procedure (see eq. (3)), step 2 is unlikely able to reach the global maxima when $D$ is a highly non-concave function.

This issue with gradient-based search is alleviated by the alternating optimization procedure: if at step 2 samples of $p_{-k}$ are stuck at local maxima, step 3 immediately decreases $D$ outputs on these samples. In other words, local maxima are constantly being eliminated. We can clearly observe this pattern in a 2D simulation of the algorithm (Figure 4).

However, it appears local maxima elimination cannot solve all the issues. As illustrated in Figure 2(b), the maximin solver could converge to a solution where $D$ has $> \frac{1}{2}$ outputs on places other than $\text{Supp}(p_k)$. Inspecting the gradient vector field in Figure 2(b), we find that by starting from $p_{-k}$ and following the gradient of $D$, $p_t$ is always "trapped" to $\text{Supp}(p_k)$. As a result, other local maxima lost the chance of being visited by $p_t$, and cannot be eliminated.

The above observation points out a straightforward solution: use a $p_{-k}$ that is distributed in the entire data space, as opposed to one that is concentrated in a subspace. For instance, when we use a *uniform distribution* in the data space as $p_{-k}$, in multiple experiments we consistently obtained $D$ solutions with no local maxima and global maxima at $\text{Supp}(p_k)$ (Figure 2(c) and Figure 5). The mathematical proof that when $p_{-k}$ is a uniform distribution we can always get such a $D$ solution is provided in Appendix M. (Note however the use of uniform distribution is not a prerequisite here; any "well distributed" data should work just as well.) The fact that these $D$ solutions don't have local maxima also means we can translate an arbitrary data point out of $\text{Supp}(p_k)$ to $\text{Supp}(p_k)$ by performing gradient ascent on $D$.

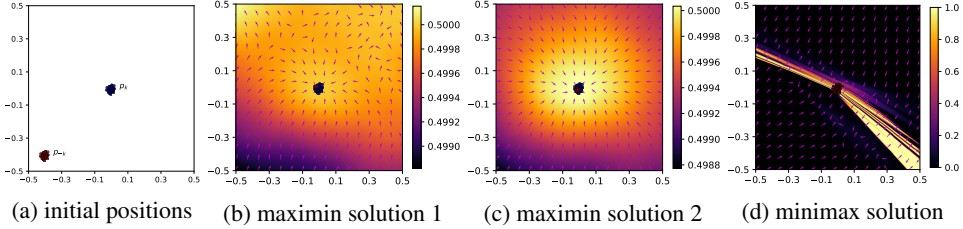

(a) initial positions     (b) maximin solution 1     (c) maximin solution 2     (d) minimax solution

Figure 2: Plots of contours and gradient vector fields of the $D$ functions (gradient vectors are normalized to have the unit length). (a) The initial positions of $p_{-k}$ and $p_k$. (b) The solution obtained by the maximin problem solver. (c) The solution obtained by the maximin problem solver when $p_{-k}$ is a uniform distribution in the data space. (d) The solution obtained by the minimax problem solver.

## 3.3 THE MINIMAX PROBLEM

---
**Algorithm 2** Minimax Problem Solver
---
1: Initialize $p_t \leftarrow p_{-k}$.
2: **repeat**
3:     Train the detector by maximizing $\mathbb{E}_{\mathbf{x} \sim p_k}[\log D(x)] + \mathbb{E}_{\mathbf{x} \sim p_t}[\log(1 - D(x))]$ (until converge).
4:     For each sample $x \in p_t$, update its value $x \leftarrow x - \lambda \frac{\nabla \log(1 - D(x))}{\|\nabla \log(1 - D(\tilde{x}))\|_2}$.
5: **until** $p_t$ convergences to $p_k$
---

The corresponding minimax game $\min_{p_t} \max_D U(D, p_t)$ has a reversed rule: player 1 first presents different $p_t$s, then for each $p_t$, player 2 determines a $D^{p_t}$ that maximizes $U$ under the considered $p_t$. Then over all the combinations of $(p_t, D^{p_t})$, player 1 chooses the combination that gives the least $U$ value. The solution of the game is analyzed in Goodfellow et al. (2014); Goodfellow (2016): the optimal strategy for player 2 is to choose a $D$ such that $U$ measures the Jensen-Shannon divergence (JSD): $U(D^*, p_t) = -\log(4) + 2 \cdot \text{JSD}(p_t \parallel p_k)$ (the actual solution of $D$ is $D^* = \frac{p_k}{p_k + p_t}$), and optimal strategy for player 1 is to choose a $p_t$ that minimizes the JSD: $p_t^* = \arg\min_{p_t \in \mathcal{M}_+^1(\mathcal{S})} \text{JSD}(p_t \parallel p_k)$. When $\text{Supp}(p_k) \subset \mathcal{S}$ (corresponding to scenario 1 in Figure 1), $p_t^*$ matches $p_k$ (JSD is zero), and $D^*$ outputs $\frac{1}{2}$ on $\text{Supp}(p_k)$. A solver (Algorithm 2) for the minimax problem is readily available by removing the "generator" from GANs' training algorithm. It is straightforward to apply GANs algorithm's convergence property to Algorithm 2: if at each step $p_t$ is updated with a sufficiently

small step of $\lambda$, and $D$ is trained to reach its optimum, then $p_t$ converges to $p_k$. In Figure 2(d) and Figure 6 we provide 2D simulation results of this algorithm.

### 3.4 THE DIFFERENCE

There are a few differences between the solutions of the maximin problem and minimax problem.

First, both $D^*$s output $\frac{1}{2}$ for Supp($p_k$). But while $D^*$ in the maximin problem outputs $\leq \frac{1}{2}$ on $\mathcal{S} \setminus \text{Supp}(p_k)$, $D^*$ in the minimax game doesn't need to be defined on $\mathcal{S} \setminus \text{Supp}(p_k)$ (Goodfellow et al., 2014). In other words, $D^*$ in the minimax problem has unpredictable values between 0 and 1 in most of the data space. We can observe this phenomenon in Figure 6. The intuition here is that in the maximin game, $p_t^*$ is decided in the second move, with the knowledge of the current $D$ value; to prevent $p_t^*$ from taking this advantage, the best strategy for player 1 is to specify $D$ outputs for the entire perturbation space. In the minimax game, on the contrary, $D^*$ is decided in the second move, with the knowledge of $p_t$, hence the player does not need to worry about $D^*$ outputs outside the supports of $p_t$ and $p_k$.

Another difference, which can also be observed from Figure 2, is that in the minimax game, $p_t^*$ exactly matches $p_k$, while in the maximin game, mass of $p_t^*$ can be any place where $D^*$ outputs $\frac{1}{2}$.

Overall we find these two formulations giving rise to different applications. The minimax formulation, which is the formulation used by GANs, is perfect for learning a generator that produces a distribution that exactly matches the target data distribution. The discriminator (the $D$ model), because of its undefined behavior in most of the data space, may not be very useful. The maximin problem, if well solved (Figure 2(c)), gives a $D$ function that models a *characteristic function* of the data distribution, and could be used to solve problems that require this feature (Section 4).

---

**Algorithm 3** Unconstrained Generative Adversarial Training

---

1: **for** $K$ in $[0, 1, \ldots, N]$ **do**
2:     **for** number of training iterations **do**
3:         Sample minibatch $m$ samples $\{x_1, \ldots, x_m\}$ from $p_k$, and $m$ samples $\{\tilde{x}_1, \ldots, \tilde{x}_m\}$ from $p_{-k}$.
4:         For each sample $\tilde{x}_i$ in $\{\tilde{x}_1, \ldots, \tilde{x}_m\}$, compute the perturbed sample $\tilde{x}_i^K$ by performing $K$ steps
        normalized steepest descent $\tilde{x}_i^{k+1} \leftarrow \tilde{x}_i^k - \gamma \frac{\nabla \log(1 - D(\tilde{x}_i^k))}{\|\nabla \log(1 - D(\tilde{x}_i^k))\|_2}$ (at initialization $\tilde{x}_i^0 \leftarrow \tilde{x}_i$).
5:         Update $D$ by maximizing $\frac{1}{m} \sum_{i=1}^{m} \left[ \log D(x_i) + \log\left(1 - D(\tilde{x}_i^K)\right) \right]$ (single step).
6:     **end for**
7: **end for**

---

## 4 APPLICATIONS

In Figure 2(c) we show when $p_{-k}$ is uniformly distributed in the data space the maximin problem solver gives us a $D$ function that has no local maxima and global maxima at the support of $p_k$. This $D$ function is very useful — we can identify at least two important applications:

- **Application 1: out-of-distribution (OOD) detection** The global maxima are at Supp($p_k$) means any inputs that have lower $D$ outputs can be correctly identified as OOD inputs.

- **Application 2: generative modeling** New samples of $p_k$ can be generated by first random sampling from the data space and then translating them to the support of $p_k$ by performing gradient ascent on $D$.

For practical applications, we have to deal with spaces of high dimensionality. We first find that with uniform noise as the $p_{-k}$ dataset, we are unable to obtain a $D$ model that is useful for detecting real OOD data (Appendix H.1). This leads us to consider using a large, diverse, real image dataset, specially ImageNet, as the $p_{-k}$ dataset. Our ablation study in Appendix H.1 confirms that larger and more diverse dataset leads to better OOD detection performances. We further use data augmentation to increase the dataset's diversity. Given these strategies, data of $p_{-k}$ could still be very sparse in a high dimensional space. In order to cover more space, we consider imposing large perturbations on $p_{-k}$ data.

**Unconstrained training** To facilitate the training of a large (potentially unlimited) perturbation, we propose Algorithm 3, an unconstrained generative adversarial training algorithm. Because of the steepest descent update rule (Line 4 in Algorithm 3, note there is no `Proj` operation here), for a given $K$, the perturbation imposed on each sample has a size that is always $\leq \lambda K$; hence the algorithm can be thought as gradually increasing the perturbation limit. We found this incremental training technique necessary for training models in high dimensional space — a phenomenon also observed by Yin et al. (2020). According to the analysis in Section 3.2, the step size $\lambda$ should be set to a sufficiently small value in order for step 4 to coverage to local maxima and step 5 to eliminate these local maxima. We observed that training is stable as long as $\lambda$ is bellow a certain threshold, and this threshold is related to input size and $D$'s architecture. In the algorithm we start $K$ from 0, which means that at the first stage $D$ is trained to discriminate between $p_k$ and $p_{-k}$; this is not critical, but we found this pre-training causes following optimizations to converge faster.

## 5 EXPERIMENTS

In this section we evaluate our method on the task of generative modeling and out-of-distribution detection. Following Kurach et al. (2018), we evaluate our method on CIFAR-10 (Krizhevsky et al., 2009), CelebA-HQ-128 (Karras et al., 2017), and LSUN Bedroom-128 (Yu et al., 2015). Details of model training, data preprocessing, and dataset statistics are provided in Appendix G.

**OOD detection evaluation** For each one of the above three datasets, we use multiple OOD datasets (see Table 6) to test a $D$ model's OOD detection performances. We further assume OOD inputs remain OOD under small $L^p$-norm perturbations. Under this assumption, we consider the problem of detecting *adversarial OOD inputs* — OOD inputs that are adversarially perturbed to cause the detection to fail, and evaluate our method under this challenging scenario. We also observe that increasing $K$ in Algorithm 3 leads to changes in $D$ performances on OOD and adversarial OOD detection. To study this phenomenon, we evaluate $D$ models trained with different $K$s using OOD inputs under various levels of perturbations. We use Outlier Exposure (OE, Hendrycks et al. (2018)) as the baseline method. The idea of OE is to use an auxiliary OOD dataset of large amount of diverse data to train the OOD detector. Because we also use a large-scale, diverse dataset (ImageNet) as the $p_{-k}$ dataset, the OE approach can be thought of a special case of Algorithm 3 when we fix $K$ to $K = 0$. We use area under the receiver operating characteristic curve (AUROC) as the performance metric (details of how AUROC and adversarial AUROC are computed are in Appendix G).

**Generative modeling evaluation** We generate new $p_k$ samples by starting from some seed images and performing gradient ascent on $D$ (Appendix G provides more details on generation). Due to the similarity between the studied approach and GANs (the former solves the maximin problem while the latter solves the minimax problem), we focus on a comparison with GANs. Kurach et al. (2018) is a large-scale study on the effects of various regularization and normalization techniques on GANs, and we compare our results with the best results obtained in their work.

### 5.1 RESULTS

**OOD detection results** Table 2 shows the average OOD detection performances of our method (see Appendix J for the complete results on individual OOD datasets). For each dataset, we train multiple $D$ models with different $K$s, and test models under various levels of perturbations [2] ($\epsilon$-test, measured by $L^2$ norm). We can observe a general pattern across all the datasets: training with a larger $K$ causes model performance on lower $\epsilon$ to decrease, a phenomenon that is also observed in other adversarial training scenarios (Madry et al., 2017; Tsipras et al., 2018). The baseline method ($K = 0$ models) becomes completely ineffective when models are exposed to adversarial OOD inputs.

On CelebA-HQ-128 and Bedroom-128 datasets our method obtains strong performances on detecting both OOD and adversarial OOD inputs. Performance on CIFAR-10 dataset is relatively low. Considering the small size (4.6MB disk space) of the default ResNet-CIFAR architecutre, we replaced it with ResNet18 which is a much larger model in terms of disk space (43MB), but only observed marginal improvements on OOD and adversarial OOD detection (Appendix H.2).

---

[2] $\epsilon$ values are based on `https://github.com/MadryLab/robustness`.

Table 2 results are based on perturbations computed using PGD attacks (Madry et al., 2017) of particular combinations of steps and step size. To verify model robustness, we use a testing strategy that is widely adopted in the ML security community: use PGD attacks of different combinations of steps and step size to test model robustness. Appendix J shows that the worst results obtained with grid search are only marginally lower than reported ones in Table 2.

In Table 3 and Table 4 we report standard and adversarial OOD detection performances of our method and several state-of-the-art methods. As is discussed earlier, there is a trade-off between standard and adversarial OOD detection performance as we increase $K$ in Algorithm 3. For this reason, we have included the performances of our model trained with different $K$s ($K = 0$ and $K = 5$.) Our method uses the ResNet18 architecture and 800 Million Tiny Images (Torralba et al., 2008) as the $p_{-k}$ dataset — this combination gives the best performance; data of other settings are provided in Appendix H.2. (We note methods in Tabel 4 that rely on axuliary data also use the 800 Million Tiny Images.) It is observed from Table 3 that state-of-the-art OOD detection methods achieves strong performances on the standard OOD detection task. However, in the adversarial OOD detection task, even a tiny perturbation of $\epsilon = 0.01$ could cause non-robust models (OE and our method with $K = 0$) to fail. Meanwhile, our method with $K = 5$ outperforms several state-of-the-art methods on SVHN and CIFAR-100 in this task.

Table 2: Out-of-distribution detection performances (see Appendix J for expanded results)

| (a) CIFAR-10 | | | | (c) CelebA-HQ-128 | | | | (d) Bedroom-128 | | |
|---|---|---|---|---|---|---|---|---|---|---|
| | $D$ model | | | | $D$ model | | | | $D$ model | |
| $\epsilon$-test | $K = 0$ | $K = 15$ | $K = 25$ | $\epsilon$-test | $K = 0$ | $K = 20$ | $K = 40$ | $\epsilon$-test | $K = 0$ | $K = 20$ | $K = 40$ |
| 0.0 | 0.974 | 0.910 | 0.860 | 0.0 | 1.000 | 1.000 | 1.000 | 0.0 | 0.996 | 0.996 | 0.970 |
| 1.0 | 0.005 | 0.784 | 0.763 | 5.0 | 0.001 | 0.998 | 0.998 | 5.0 | 0.000 | 0.976 | 0.938 |
| 2.0 | 0.000 | 0.519 | 0.613 | 10.0 | 0.000 | 0.964 | 0.988 | 10.0 | 0.000 | 0.815 | 0.837 |

Table 3: Standard OOD detection performances (AUROC scores) when the in-distribution dataset is CIFAR-10 and OOD samples are not perturbed. Performance data is collected from referenced papers in the table; when there is a discrepancy we use the best reported result. Details about the iSUN, LSUN (resize), and TinyImageNet (resize) datasets can be found at Liang et al. (2017).

| Method | OOD dataset (no perturbation) | | | | | | |
|---|---|---|---|---|---|---|---|
| | Uniform | Gaussian | SVHN | CIFAR-100 | iSUN | LSUN(resize) | TinyImageNet (resize) |
| Softmax (Hendrycks & Gimpel, 2016) | 96.5 | 97.5 | 89.9 | 86.4 | 91.0 | 91.0 | 91.0 |
| ODIN (Liang et al., 2017) | 99 | 100 | 96.7 | 87.5 | 94.0 | 94.1 | 94.0 |
| Mahalanobis (Lee et al., 2018) | 100 | N/A | 99.1 | 88.2 | 99.5 | 99.7 | 99.5 |
| OE (Hendrycks et al., 2018) | 98.7 | 99.3 | 98.8 | 95.3 | 98.5 | 98.94 | N/A |
| Gram Matrices (Sastry & Oore, 2019) | N/A | 100 | 99.5 | 79.0 | 99.8 | 99.9 | 99.7 |
| Energy-based (Liu et al., 2020) | N/A | N/A | 99.4 | N/A | 99.33 | 99.39 | N/A |
| Likelihood ratios (Ren et al., 2019) | N/A | N/A | 88.8 | N/A | N/A | N/A | N/A |
| WAIC (Choi et al., 2018) | 100 | 100 | 100 | N/A | N/A | N/A | 95.6 |
| CCU (Meinke & Hein, 2019) | 100 | N/A | 97.1 | 93.0 | N/A | N/A | N/A |
| ACET (Hein et al., 2019) | 99.7 | N/A | 92.4 | 90.7 | N/A | N/A | N/A |
| GOOD (Bitterwolf et al., 2020) | 99.5 | N/A | 97.1 | 92.9 | N/A | N/A | N/A |
| Ours ($K = 0$) | 99.5 | 99.8 | 99.6 | 94.1 | 99.5 | 99.5 | 98.7 |
| Ours ($K = 5$) | 99.6 | 99.9 | 97.4 | 91.5 | 98.5 | 98.9 | 96.2 |

**Image generation results** In Figure 3 shows samples generated by our method. In general we find our results to be more recognizable than GANs' results in Figure 10. We observed that the quality of generated images could be affected by the type of seed images (Appendix K.2), and increasing $K$ in Algorithm 3 generally leads to better generations (Appendix K.1). In Figure 12 we demonstrate the method's application to face retouching. The fact that generated images are not realistic and have various artifacts suggest that the maximin problem is not well solved. This might be due to the model not being exposed to enough $p_{-k}$ data (consider increase the number of iterations in the inner loop of Algorithm 3), or the limitation of model architecture or capacity.

While GANs has various training stability issues, we found Algorithm 3 to be as stable as ordinary supervised training. The only failure mode (gradient ascent on $D$ results in noisy images) that we observed is caused by $\lambda$ being too large (Appendix I).

**Limitation** As our method uses gradient ascent which is susceptible to local maxima to generate samples, it tends to produce similar samples if the seed samples are not diverse enough. In practice,

Table 4: Adversarial OOD detection performances (AUROC scores) when the in-distribution dataset is CIFAR-10. Performance data of methods other than ours is collected from Bitterwolf et al. (2020). The results of our method are based on an PGD attack of steps 100 and step size 0.002. We also used the full datasets to run the test, as opposed to 1000 samples per dataset as used by Bitterwolf et al. (2020). For results using 1000 samples and under different attack configurations including one with random restarts see Table 30.

| Method | OOD dataset (with an $L^\infty$ perturbation of $\epsilon = 0.01$) | | | |
| --- | --- | --- | --- | --- |
| | Uniform Noise | Gaussian Noise | SVHN | CIFAR-100 |
| OE (Hendrycks et al., 2018) | 75.7 | N/A | 3.7 | 11.0 |
| CCU (Meinke & Hein, 2019) | **100** | N/A | 14.8 | 23.3 |
| ACET (Hein et al., 2019) | 98.9 | N/A | 88.0 | 74.5 |
| GOOD (Bitterwolf et al., 2020) | 99.5 | N/A | 58.9 | 54.7 |
| Ours ($K = 0$) | 97.8 | 22 | 1.0 | 7.1 |
| Ours ($K = 5$) | 99.0 | 99.1 | **91.8** | **81.8** |

due to the numerical algorithm or mini-batch training, we are more likely to get a $D$ solution that has local maxima. In that case, by performing gradient ascent on $D$ seed samples that are not diverse enough could be trapped in the same local maximum point. This is likey the case in Figure 3 where we find several face images that are quite similar to each other. On the other hand, in the less likely situation where the $D$ solution has no local maxima (e.g., Figure 2(b) and Figure 2(c)), all the seed samples could be concentrated in a few maxima points on support of the $p_k$ distribution. While the problem in the latter case seems more severe, it could be mitigated by properly constraining the number of steps and step size when performing gradient ascent on $D$.

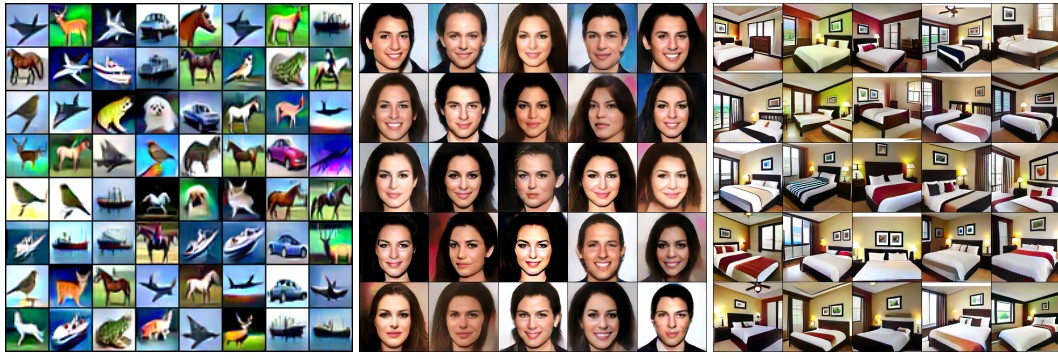

Figure 3: *Uncurated* samples generated by our method; GANs results is in Figure 10. Seed images used to generated these results are in Figure 11. The training time for models used to produce these generations are in Table 7.

## 6 CONCLUSIONS AND FUTURE WORK

In this paper we analyzed the optimal solutions of the GAT training objective and the convergence property of the training algorithm. The analysis of optimal solutions justifies the application of the GAT method to training robust predictive models. We made a comparative analysis of the maximin formulation and minimax formulation that are respectively employed by GAT and GANs. Guided by these theoretical results, we designed an unconstrained GAT algorithm, and evaluated it on the task of image generation and adversarial out-of-distribution detection. The competitive performance and training stability of the algorithm suggest that the studied approach could serve as a new tool for content creation, although we believe its performance could be further improved by optimizing hyperparameters and model architectures. Out-of-distribution detection results indicate that an OOD detection model's robustness could be improved by training the model against an adversary equipped with large-scale, diverse OOD data. The future work includes scaling up the training for larger images and high-capacity models, and extending the method's application to sequential and tabular data.

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

## A    RELATED WORK ON OUT-OF-DISTRIBUTION DETECTION

Out-of-distribution (OOD) detection, also known as novelty detection, or anomaly detection, deals with the problem of identifying novel, or unusual, data from within a dataset. OOD detection has gained much research attention due to its practical importance in safety-critical applications and changeling nature. A comprehensive review of classical OOD detection methods can be found at Pimentel et al. (2014).

A recent surge of research interests in this topic is due to the emergence of deep generative models. Such models (specifically explicit density models (Goodfellow, 2016)) estimate the generative probability density function of the data, and should serve as an ideal candidate for OOD detection. However, it was observed (Kirichenko et al., 2020; Nalisnick et al., 2018; Shafaei et al., 2018; Hendrycks et al., 2018) that several state-of-the-art deep generative models, including Glow (Kingma & Dhariwal, 2018), PixelCNN (Oord et al., 2016), PixelCNN++ (Salimans et al.), VAEs (Kingma, 2013; Rezende et al., 2014), and RealNVP flow model (Dinh et al., 2016) tend to assign higher likelihood to OOD inputs than they do to in-distribution inputs. Despite this challenge, several recent works (Ren et al., 2019; Choi et al., 2018; Nalisnick et al., 2019; Kirichenko et al., 2020; Serrà et al., 2019; Song et al., 2019; Huang et al., 2019; Daxberger & Hernández-Lobato, 2019) investigated the issue and successfully applied deep generative models to OOD detection.

There is also a plethora of OOD detection methods (Hendrycks & Gimpel, 2016; Lee et al., 2018; Liang et al., 2017; Sastry & Oore, 2019; Quintanilha et al., 2018; Abdelzad et al., 2019; Chen et al., 2018; Malinin & Gales, 2018) that make use of statistics computed from the predictions or intermediate activations of standard classifiers train on in-distribution data. To name a few, Lee et al. (2018) fit class conditional Gaussian distributions using multiple levels of activations of the classifier, and use Mahalanobis distance to compute confidence scores to identify OOD inputs. The ODIN method (Liang et al., 2017) improves the effectiveness of a softmax score based detection approach by using temperature scaling and adding small perturbations to the input. Sastry & Oore (2019) make use of gram matrices computed from the classifier's intermediate activations to identify OOD inputs.

Another branch of work utilize various alternative training strategies (Liu et al., 2020; Lee et al., 2017; Hendrycks et al., 2018; 2019; DeVries & Taylor, 2018; Shalev et al., 2018; Vernekar et al., 2019; Yu & Aizawa, 2019; Golan & El-Yaniv, 2018). A notable example is the Outlier Exposure (OE) method developed by (Hendrycks et al., 2018). OE works by training the OOD detector against a large, diverse out-of-distribution dataset, and has been widely adopted as a baseline method.

While methods based on generative models and standard classifiers yield high performances on naturally-occurring OOD inputs, several such methods have been shown (Meinke & Hein, 2019; Bitterwolf et al., 2020) to be vulnerable to adversarial manipulation of the OOD inputs. This should come as no surprise as both generative models and standard classifiers themselves are vulnerable to adversarial attacks (Kos et al., 2018; Szegedy et al., 2013). Given the above limitation of current approaches, a recent trend considers the worst-case scenario for OOD detection (Hein et al., 2019; Sehwag et al., 2019; Meinke & Hein, 2019; Bitterwolf et al., 2020). The Adversarial Confidence Enhanced Training (ACET) method proposed by Hein et al. (2019) use adversarial training Madry et al. (2017) on OOD inputs to improve detection robustness. Meinke & Hein (2019) uses a density estimator to provide guarantees on the maximal confidence around $L^2$ ball for uniform noise. Bitterwolf et al. (2020) use interval-bound-propagation (IBP) to certificate worst case guarantees for general OOD inputs under a $L^\infty$ threat model. In the same spirit as Hein et al. (2019), our detection method employs adversarial training on OOD inputs to induce robustness. The difference is that our method uses the GAT objective with a optimal solution which naturally solves the robust OOD detection problem, while the optimal solution of the objective used by Hein et al. (2019), which is essentially a multiple class classification objective with an extra term on OOD inputs, is unclear.

## B    MAXIMIN AND MINIMAX PROBLEMS IN GAME THEORY

In game theory, two-player zero-sum game is a mathematical representation of a situation in which one player's gain is balanced by another player's loss. Such a game is described by its *payoff function* $f : \mathbb{R}^{p+q} \to \mathbb{R}$, which represents the amount of payment that one player (player 1) makes to the

other player (player 2). The goal of player 1 is to choose a strategy $u \in \mathbb{R}^p$ such that the payoff is minimized, while the goal of player 2 is to choose a strategy $u \in \mathbb{R}^q$ such that the payoff is maximized. The best strategies for both players, and the resulting payoff, depending on the order of play, could be solved via $\min_u \max_v f(u, v)$ or $\max_v \min_u f(u, v)$.

In the minimax game $\min_u \max_v$, player 1 makes the first move. Player 2, after learning that player 1 has made the move $u$, will choose a $v$ to maximize $f(u, v)$, which results in a payoff of $\max_v f(u, v)$. Player 1, who is informed of player 2's strategy, will choose a $u$ such that the worse case payoff $\max_v f(u, v)$ is minimized, which results in a payoff of $\min_u \max_v f(u, v)$.

In the maximin game $\max_v \min_u$, the order of play is reversed. Player 2 makes the first move, and then player 1 minimizes the payoff by choosing $u = \arg\min_u f(u, v)$. Player 2 knows that player 1 will follow this strategy and will choose a $v$ such that the worse case payoff $\min_u f(u, v)$ is maximized, which results in a payoff of $\max_v \min_u f(u, v)$.

The payoff $\min_u \max_v f(u, v)$ is always greater or equal to $\max_v \min_u f(u, v)$. This difference can be intuitively understood as the result of player 2's extra knowledge gained by taking the second move. According to the minimax theorem (Neumann, 1928), when $f$ is a continuous function that is concave-convex (i.e., for each $v$, $f(u, v)$ is a convex function of $u$, and for each $u$, $f(u, v)$ is a concave function of $v$)), these two quantities are equal. We refer the reader to Boyd et al. (2004) (§5.4.3, §10.3.4) for more details on this topic.

## C  A DEMONSTRATION ON HOW TO SOLVE A MAXIMIN PROBLEM

Without a game theory interpolation, in Table 5 we present a minimal example demonstrating how to solve a maximin problem $\max_v \min_u f(u, v)$, with $f : \mathbb{R}^{p+q} \to \mathbb{R}$, $u \in \mathbb{R}^p$, and $v \in \mathbb{R}^q$. In this example, $u$ has three values $u_0$, $u_1$, $u_2$, and $v$ has two values: $v_0$, $v_1$. To solve the maximin problem we first solve the inner minimization for each value of $v$. As an example, when we fix $v$ to $v_0$, we solve the inner problem by choosing the $u$ that when combined with $v_0$, yields the *lowest* $f$ value. We do the same computation for $v_1$ and we have solved the inner problem. We then move forward to the outer problem by choosing from the above two solutions (red boxes) the one with the *highest* $f$ value (the green box).

Table 5: A minimal example demonstrating how to solve a maximin problem. The solutions for the inner problem for each value of $v$ are labeled as red, and the final solution is highlighted as green.

|       | $u_0$       | $u_1$       | $u_2$       |   |       | $u_0$       | $u_1$       | $u_2$       |
|-------|-------------|-------------|-------------|---|-------|-------------|-------------|-------------|
| $v_0$ | $f(u_0, v_0)$ | $f(u_0, v_1)$ | $f(u_0, v_2)$ |   | $v_0$ | $f(u_0, v_0)$ | $f(u_0, v_1)$ | $f(u_0, v_2)$ |
| $v_1$ | $f(u_1, v_0)$ | $f(u_1, v_1)$ | $f(u_1, v_2)$ |   | $v_1$ | $f(u_1, v_0)$ | $f(u_1, v_1)$ | $f(u_1, v_2)$ |

## D  MATHEMATICAL ANALYSIS OF OPTIMAL SOLUTIONS OF THE MAXIMIN PROBLEM

Recall that the support of $p_t$ can be any subset of the pertubation space $\mathcal{S}$ and that $U(D, p_t) = \int p_k(x) \log D(x)) dx + \int p_t(x) \log(1 - D(x)) dx$. For convenience, we define the contour set inside $\mathcal{S}$ of $D$ at $\alpha$ as $C_\alpha^D := \{x \in \mathcal{S} : D(x) = \alpha\}$, the region of $\text{Supp}(p_k)$ that is outside of $\mathcal{S}$ as $\Omega_{ko} := \text{Supp}(p_k) \setminus \mathcal{S}$ and the region of $\text{Supp}(p_k)$ that is in $\mathcal{S}$ as $\Omega_{ki} := \text{Supp}(p_k) \cap \mathcal{S}$. Note that $\text{Supp}(p_k) = \Omega_{ko} \cup \Omega_{ki}$. For a fixed $D$ and let $\alpha_o = \max_{\Omega_{ko}} D$ and $\alpha_S = \max_{\mathcal{S}} D$. It is easy to check that $U$ is minimized when $\text{Supp}(p_t)$ lies in the contour set $C_{\alpha_S}^D$. Let $p_t^*$ be a distribution such that $\text{Supp}(p_t^*) \subset C_{\alpha_S}^D$. By direct computation we have that

$$U(D, p_t^*) = \int_{\Omega_{ko}} p_k(x) \log D(x) dx + \int_{\Omega_{ki}} p_k(x) \log D(x) dx + \log(1 - \alpha_S)$$

$$\leq (\int_{\Omega_{ko}} p_k)(\log \alpha_k) + (\int_{\Omega_{ki}} p_k)(\log \alpha_S) + \log(1 - \alpha_S)$$

$$\leq 0 + \beta_{ki} \log \frac{\beta_{ki}}{1 + \beta_{ki}} + \log \frac{1}{1 + \beta_{ki}},$$

where $\beta_\epsilon = \int_{\Omega_{ki}} p_k$. Note here we have used the fact the the function $f(y) = a \log y + b \log(1-y)$ achieves its maximum at $y = \frac{a}{a+b}$. It is not difficult to see that the above inequality becomes an equality when

$$D(x) = \begin{cases} \alpha_k & x \in \Omega_{ko} \\ \alpha_S & x \in \Omega_{ki} \\ \leq \alpha_S & x \in \mathcal{S} \setminus \text{Supp}(p_k) \end{cases}, \tag{6}$$

where $\alpha_k = 1$ and $\alpha_S = \frac{\beta_{ki}}{1+\beta_{ki}}$. Note that $D$ does not need to be defined outside $\mathcal{S} \cup \text{Supp}(p_k)$.

**Scenario 1**   Here we deal with the case when $\epsilon$ is large enough such that $\text{Supp}(p_k) \subset \mathcal{S}$, in which case $\Omega_{ko} = \emptyset$, $\Omega_{ki} = \text{Supp}(p_k)$ and $\alpha_S = \frac{1}{2}$. Hence by the above analysis one can check that $U$ achieves its optimum when $D \equiv \alpha_S = \frac{1}{2}$ on $\text{Supp}(p_k)$ and $D \leq \frac{1}{2}$ on $\mathcal{S} \setminus \text{Supp}(p_k)$. In summary, the maximin problem achieves its optimum when $D$ outputs $\frac{1}{2}$ on the support of $p_k$ and and values less or equal to $\frac{1}{2}$ on samples outside the support of $p_k$ but in $\mathcal{S}$.

**Scenario 2**   Here we deal with the case when $\epsilon$ is small enough such that the $\mathcal{S} \cap \text{Supp}(p_k) = \emptyset$, in which case $\Omega_{ko} = \text{Supp}(p_k)$, $\Omega_{ki} = \emptyset$ and $\alpha_S = 0$. Hence $U$ achieves its optimum when $D \equiv 1$ on $\text{Supp}(p_k)$ and $D \equiv 0$ on $\mathcal{S}$. In summary, the maximin problem achieves its optimum when $D$ outputs $1$ on the support of $p_k$ and zero on the the perturbation space $\mathcal{S}$.

**Scenario 3**   Here we deal with the case when $\mathcal{S} \cap \text{Supp}(p_k) \neq \emptyset$ and $\text{Supp}(p_k) \not\subset \mathcal{S}$. In summary, the maximin problem achieves its optimum when $D$ outputs $1$ on the set of samples inside the support of $p_k$ but outside of the perturbation space $\mathcal{S}$ and $\frac{\beta_{ki}}{1+\beta_{ki}}$ on the set of samples that are in the intersection of the support of $p_k$ and $\mathcal{S}$ and values less or equal to $\frac{\beta_{ki}}{1+\beta_{ki}}$ on $\mathcal{S}$.

**Remark**   The first two cases can be seen as the special cases of the third one.

## E   SCENARIO 2 DISCUSSION

In robust machine learning literature, it's common to consider a very small value for $\epsilon$. For instance, one of the most commonly used limit for training $L^\infty$ robust models is $\epsilon = 8/255$ ($L^\infty$ norm). A perturbation space characterized by a small limit can be thought as a semantic-preserving space: translating a sample inside the space doesn't change the sample's underlying label/class membership. A small perturbation limit corresponds to scenario 2, which is also the focus of Yin et al. (2020). We can define robust models as models that output consistent predictions for inputs under semantic-preserving transformations. In this sense, the optimal $D$ for scenario 2 is a robust detector, as it always outputs 0 for the perturbation space. However, the limitation of training against a small $\epsilon$ is obvious: because optimal $D$'s outputs outside $\mathcal{S} \cup \text{Supp}(p_k)$ are unspecified, any semantic-preserving operation that has a perturbation that goes beyond $\mathcal{S}$ can result in a high $D$ output, thereby fools the detection. The above analysis suggests that for predictive models based on the generative adversarial training method, their robustness can be improved by training against a larger perturbation space.

## F   ALGORITHM 1 CONVERGENCE

### F.1   STEP 3 ALWAYS DECREASE $\beta$

We assume when $\alpha < \beta$, $D$ has a single global maximum point (i.e., $|B| = 1$).

**Lemma 1.** *If $\alpha < \beta$ and $|B| = 1$, Algorithm 1 always decreases $\beta$.*

*Proof.* Let $\beta := \max_{\text{Supp}(p_k)} D$, $C := \{x \in \text{Supp}(p_k) : D(x) < \beta\}$, and $\gamma := \max_C D$. (Same as the proof for Proposition 2, we consider the case of $\beta, \gamma > \frac{1}{2}$.) Going back to Algorithm 1, step 2 moves mass of $p_{-k}$ to $B$. Since $B$ has only one element, the mass is concentrated on this single point. Step 3's optimization causes $\gamma$ to increase and $\beta$ to decrease. The intuition here is that if step 3's update is small enough, these two values will meet in an intermediate point. Let the resulting values

be $\gamma_1$ and $\beta_1$. If step 3's update on $D$ is sufficiently small such that $\gamma_1 - \gamma < \beta - \gamma$, then we have $\max\{\beta_1, \gamma_1\} < \beta$ — the maximum value of $D$ on $\text{Supp}(p_k)$ has decreased. $\qquad\square$

## F.2 THE EFFECTS OF ALTERNATING OPTIMIZATION

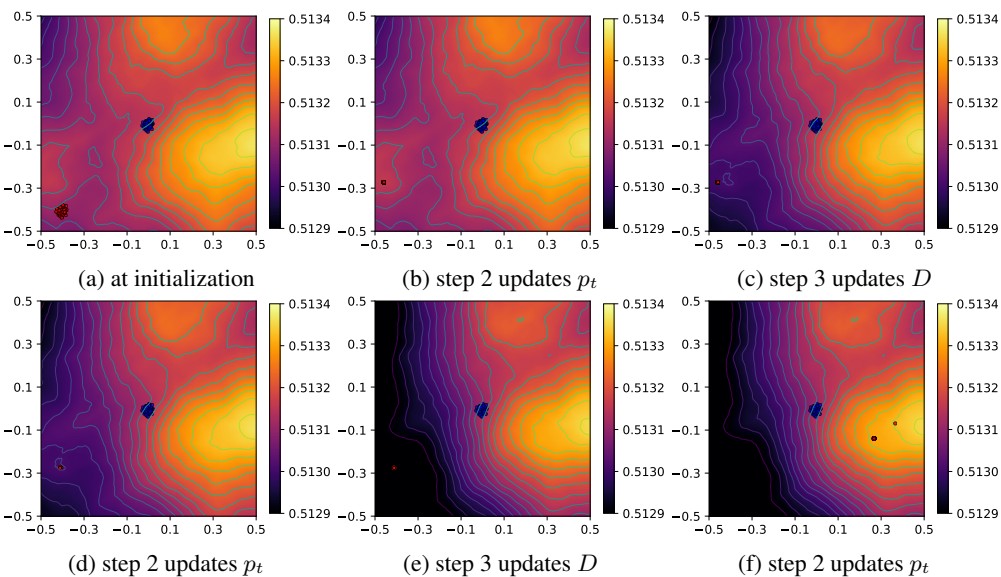

Figure 4: The results of $p_t$ and $D$ in the first few iterations of a 2D simulation of Algorithm 1. Step 2 solves the inner minimization, causes support of $p_t$ (red points) to be concentrated in local maxima points. Step 3 update $D$ by increasing its outputs on the support of $p_k$ and decreasing its outputs on the support of $p_t$, causes local maxima to be suppressed.

### F.3 MORE 2D SIMULATION RESULTS

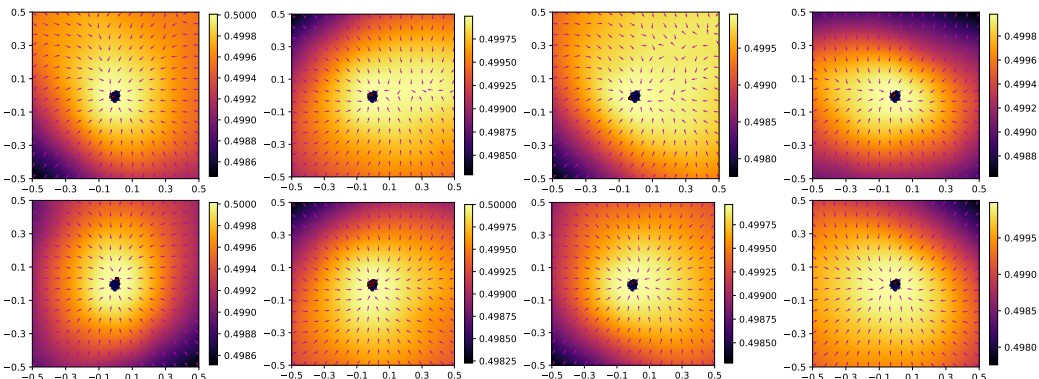

Figure 5: Solutions obtained by the *maximin* problem solver (Algorithm 1) with different initializations of $D$. First row are results when $p_{-k}$ is at bottom left (see Figure 2), and second row are results when $p_{-k}$ are uniform distributions.

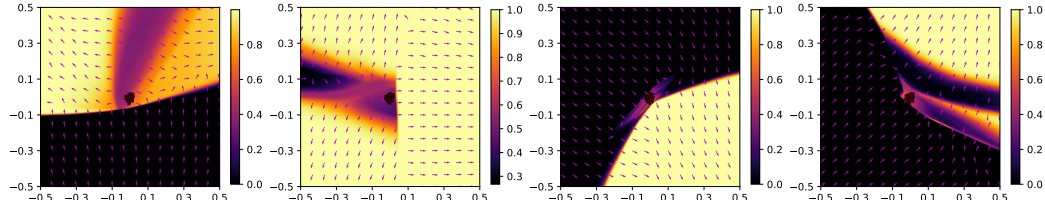

Figure 6: Solutions obtained by the *minimax* problem solver (Algorithm 2) with different initializations of $D$. Note that in all cases $p_t$ (red distribution) matches $p_k$ (blue distribution), but $D$ has unpredictable outputs on $\mathcal{X} \setminus \text{Supp}(p_k)$. The initial position of the red distribution is in bottom left corner (see Figure 4(a)).

# G  EXPERIMENTAL SETUP

**Model training**     We use Algorithm 3 to train $D$ models. Depending on the studied dataset, $p_k$ is set to one of the above three datasets, but we always use ImageNet (with data augmentation) as the $p_{-k}$ dataset (images are resized to $32 \times 32 \times 3$ or $128 \times 128 \times 3$ depending on $p_k$ dataset resolution). We also use the same $D$ architectures and batch size as Kurach et al. (2018) (see Appendix G for more details of $D$ architectures).

Following Yin et al. (2020), we use a pretrained $D$ model to bootstrap optimization: for CIFAR-10 the $D$ model is pretrained on the CIFAR-10 classification task, and for other datasets it is pretrained on the ImageNet classification task (Russakovsky et al., 2015). For all the datasets, the $D$ update step in Algorithm 3 is performed using a SGD optimizer with momentum of 0.9. The learning rate of the optimizer for CIFAR-10 datast is 0.0005, for CelebA-HQ-128 is 0.001, and for Bedroom-128 is 0.0025. The $\lambda$ value is set to 0.1 for CIFAR-10, and 0.6 for CelebA-HQ-128 and Bedroom-128. For all the trainings we use a batch size of 64, same as Kurach et al. (2018). The training follows standard supervised training, and doesn't use any regularization or normalization.

**Dataset preprocessing and statistics**     CelebA-HQ-128 dataset is downloaded from Manna (2020). The Bedroom-128 dataset is created from the corresponding LSUN dataset by center cropping the images using a square and then resizing to $128 \times 128$. CIFAR-10 has 60K training images and 10K test images. We manually split CelebA-HQ-128 into a training split of 27K images and a test split of 3K images, and Bedroom-128 into a training split of 300K images and a test split of 3K images.

**Detector architecture**     Following Kurach et al. (2018), we use tow network architecture for the experiments. For CIFAR-10 task we use the "ResNet-CIFAR" architecture (Kurach et al. (2018) Table 7a). The architecture has 4 customized ResBlocks and takes 4.6MB disk space. For other $128 \times 128 \times 3$ datasets, we use the "ResNet19 discriminator" architecture (Kurach et al. (2018) Table 5a). The architecture has 6 customized ResBlocks and takes 60MB disk space

**Image generation details**     We generate a new sample of $p_k$ by starting from some seed sample and performing gradient ascent on $D$ using the update rule in eq. (3). In this case, the seed sample is supposed to be out of the distribution of $p_k$. When using the update rule in eq. (3), we need to specify the step size $\lambda$, the number of steps, and $\epsilon$ for the `Proj` operation. We use the following configurations for generation:

- For CIFAR-10 generation (Figure 3(a), Figure 13, and Figure 16), we use step size 0.1, steps 200, and $\epsilon = 15$.

- For CelebA-HQ-128 generation (Figure 3(b), Figure 14, and Figure 17, we use step size 1.2, steps 100, and $\epsilon = 40$.

- For CelebA-HQ-128 generation (Figure 3(c), Figure 15, and Figure 18, we use step size 0.8, steps 400, and $\epsilon = 70$.

We note different configurations could lead to different generation results.

**AUROC computation**     AUROC is a metric that measures a discriminative model's ability to separate two sets of data. To compute the AUROC score of a trained $D$ model for a given $p_k$ and $p_{\text{OOD}}$ dataset, we first use the $D$ model to get the logit outputs of samples of these two datasets, and then use scikit-learn (Pedregosa et al., 2011)'s "sklearn.metrics.auc" function to compute the score (with samples in $p_k$ labeled as 1s and samples in $p_{\text{OOD}}$ labeled as 0s). We always use the test splits of the $p_{\text{OOD}}$ and $p_k$ datasets to do the above calculation.

**Adversarial AUROC computation**     To compute the adversarial AUROC score of a $D$ model for a given $p_{\text{OOD}}$ and $p_k$ dataset, we first compute an adversarial OOD dataset $p'_{\text{OOD}}$ by taking samples from $p_{\text{OOD}}$ and performing $L^2$-based PDG attack (Madry et al., 2017) against the $D$ model. We then compute the Adversarial AUROC score by computing the AUROC score on the $p'_{\text{OOD}}$ and $p_k$ datasets. Same as the AUROC computation, we always use the test splits of the $p_{\text{OOD}}$ and $p_k$ to do the above calculation.

Table 6: In-distribution dataset and corresponding out-of-distribution datasets (images of OOD datasets are resized to the image size of the corresponding in-distribution dataset)

| In-distribution dataset ($p_k$) | Out-of-distribution datasets ($p_{OOD}$) |
|---|---|
| CIFAR-10 | Gaussian noise, Uniform noise, SVHN (Netzer et al., 2011), CIFAR-100 (Krizhevsky et al., 2009), ImageNet, CelebA-HQ-128, Bedroom-128 |
| CelebA-HQ-128 | Gaussian noise, Uniform noise, SVHN, CIFAR-100, ImageNet, CIFAR-10, Bedroom-128 |
| Bedroom-128 | Gaussian noise, Uniform noise, SVHN, CIFAR-100, ImageNet, CelebA-HQ-128, CIFAR-10 |

Table 7: Model training time.

| Model | Training Time |
|---|---|
| CIFAR-10 $K = 40$ model | 2 days 23 hours (1 2080Ti GPU) |
| CelebA-HQ-128 $K = 80$ model | 7 days 12 hours (2 2080Ti GPUs) |
| Bedroom-128 $K = 55$ model | 14 days 15 hours (2 20280Ti GPUs) |

Table 8: FID scores

| | CIFAR-10 | CELEBA-HQ-128 | LSUN-BEDROOM-128 |
|---|---|---|---|
| Our method | 60.79 | 83.01 | 56.86 |
| GANs (Kurach et al., 2018) | 22.7 | 24.7 | 40.4 |

# H    ABLATION STUDY

## H.1    UNIFORM NOISE AND DATA DIVERSITY

In this ablation study, we use CIFAR-10 class 0 data as the target data distribution dataset $p_k$, and train models with different $p_{-k}$ datasets.

It is observed in Table 11 and 12 that when $p_{-k}$ is uniform noise, the $D$ models only developed capability for identifying uniform noise and Gaussian noise as OOD inputs. This result seems to contradict the mathematical analysis in Appendix M which says that with a uniform distribution as $p_{-k}$ a $D$ function useful for detecting any kind of OOD inputs could be obtained. According to the manifold hypothesis, real image data lie on lower-dimensional manifolds embedded within the high-dimensional space. While in contrast, the uniform noise is highly concentrated on the surface of the unit $d$-cube[3] in a high dimensional space $[0, 1]^d$. Our conjecture is that due to these geometry properties, in terms of Euclidean distance, real image data is close to each other while uniform noise live far away from the real data. As a result, uniform noise is much less data efficient than real data for training OOD detection models, and a much larger number of inner iterations and $K$ value in Algorithm 3 may be needed to reach a satisfying detection performance.

For real image data experiments, we respectively use ImageNet and a CIFAR-10 subset consisting of CIFAR-10 data from class 1 to class 9 as the $p_{-k}$ dataset. ImageNet is a considerably much larger and more diverse dataset than the CIFAR-10 dataset, and it is seen from Table 9 and Table 10 that the model training against Imagenet performs much better on OOD and adversarial OOD detection than the model trained against the CIFAR-10 subset.

Table 9: Average OOD performance (AUROC scores) on CIFAR10 class 0 data. ($p_k$ = CIFAR-10 class 0, and $p_{-k}$ = ImageNet).

|  | $D$ model | | |
| --- | --- | --- | --- |
| $\epsilon$-test | $K = 0$ | $K = 15$ | $K = 25$ |
| 0.0 | 0.9940 | 0.9872 | 0.9758 |
| 1.0 | 0.1352 | 0.9330 | 0.9325 |
| 2.0 | 0.0084 | 0.6863 | 0.8284 |

Table 10: Average OOD detection performance (AUROC scores) on CIFAR-10 class 0 data. ($p_k$ = CIFAR-10 class 0, and $p_{-k}$ = CIFAR-10 class 1 - class 9).

|  | $D$ model | | |
| --- | --- | --- | --- |
| $\epsilon$-test | $K = 0$ | $K = 15$ | $K = 25$ |
| 0.0 | 0.9790 | 0.9699 | 0.9477 |
| 1.0 | 0.0985 | 0.8612 | 0.8732 |
| 2.0 | 0.0004 | 0.5023 | 0.6979 |

Table 11: OOD detection performance (AUROC scores) of $K = 0$ model on CIFAR-10 class 0 data ($p_k$ = CIFAR-10 class 0, and $p_{-k}$ = uniform noise).

| $\epsilon$-test | Gaussian noise | Uniform noise | ImageNet | Bedroom | SVHN | CelebAHQ | CIFAR100 | mean |
| --- | --- | --- | --- | --- | --- | --- | --- | --- |
| 0.0 | 1.0000 | 1.0000 | 0.5859 | 0.5791 | 0.5801 | 0.5235 | 0.5499 | 0.6884 |
| 1.0 | 1.0000 | 1.0000 | 0.5161 | 0.5028 | 0.5141 | 0.4510 | 0.4816 | 0.6379 |

---

[3]The volume of the unit $d$-cube shrunk by some small epsilon in each dimension is given by $V = (1 - 2\epsilon)^d$. This quantity quickly approaches 0 as d increases.

Table 12: OOD detection performance (AUROC scores) of $K = 15$ model on CIFAR-10 class 0 data. ($p_k$ = CIFAR-10 class 0, and $p_{-k}$ = uniform noise).

| $\epsilon$-test | Gaussian noise | Uniform noise | ImageNet | Bedroom | SVHN | CelebAHQ | CIFAR100 | mean |
|---|---|---|---|---|---|---|---|---|
| 0.0 | 1.0000 | 1.0000 | 0.5772 | 0.5760 | 0.5711 | 0.5139 | 0.5435 | 0.6831 |
| 1.0 | 1.0000 | 1.0000 | 0.5063 | 0.4984 | 0.5043 | 0.4406 | 0.4738 | 0.6319 |

## H.2 EFFECTS OF MODEL CAPACITY

Table 13: Adversarial OOD detection performances (AUROC scores) of our method trained with different model architectures and $p_{-k}$ datasets. The in-distribution dataset is CIFAR-10. The definition of ResNet18 can be found at `https://github.com/MadryLab/robustness`.

| Method | OOD dataset (with an $L^\infty$ perturbation of $\epsilon = 0.01$) | | | |
|---|---|---|---|---|
| | Uniform Noise | Gaussian Noise | SVHN | CIFAR-100 |
| ResNet-CIFAR, $p_{-k}$ = Imagenet ($K = 5$) | 100 | 100 | 89.2 | 74.0 |
| ResNet-CIFAR, $p_{-k}$ = 800 Million Tiny Images ($K = 5$) | 99.9 | 99.9 | 90.8 | 78.4 |
| ResNet18, $p_{-k}$ = 800 Million Tiny Images ($K = 5$) | 99.0 | 99.1 | 91.8 | 81.8 |

Table 14: Standard OOD detection performances (AUROC scores) of our method trained with different model architectures and $p_{-k}$ datasets. The in-distribution dataset is CIFAR-10.

| Method | OOD dataset (no perturbation) | | | |
|---|---|---|---|---|
| | Uniform Noise | Gaussian Noise | SVHN | CIFAR-100 |
| ResNet-CIFAR, $p_{-k}$ = Imagenet ($K = 5$) | 100 | 100 | 96.1 | 87.0 |
| ResNet-CIFAR, $p_{-k}$ = 800 Million Tiny Images ($K = 5$) | 100 | 100 | 97.0 | 89.1 |
| ResNet18, $p_{-k}$ = 800 Million Tiny Images ($K = 5$) | 99.6 | 100 | 97.4 | 91.5 |

## I  FAILURE MODE DIAGNOSIS

We observe that in Algorithm 3, if $\lambda$ is set to a too large value, the algorithm fails to learn a $D$ that is useful for image generation. In this section we discuss the training dynamics of the case of an appropriate $\lambda$ value and the case of $\lambda$ being too large.

$\lambda$ **is small enough**   In Algorithm 3 as we increase $K$, $p_t$ gradually converges to $p_k$. In this process it becomes increasingly difficulty for the $D$ model to differentiate these two distributions. This phenomenon can be observed in Figure 7 : the training loss (binary cross-entropy loss) of the $D$ model becomes larger and larger (left subfigure), and eventually these two distributions become indistinguishable (AUROC $\approx 0.5$, middle subfigure). From the right subfigure we can see that $D$'s performance on $p_{-k}$ vs. $p_k$ is also affected by the increase in $K$ value.

$\lambda$ **is too large**   The failure mode caused by $\lambda$ being too large is easy to identify (Figure 8): the training loss quickly decreases to 0 as $K$ increases (left subfigure), $p_t$ and $p_k$ become perfectly separable (middle subfigure), and $D$ model becomes unable to separate $p_{-k}$ from $p_k$(right subfigure).

In general, with a small enough $\lambda$ value, an increase of sample quality could be expected after model is trained with a larger $K$. This is the case when $\lambda$ is 0.1, but not when it is 0.6 (Figure 9).

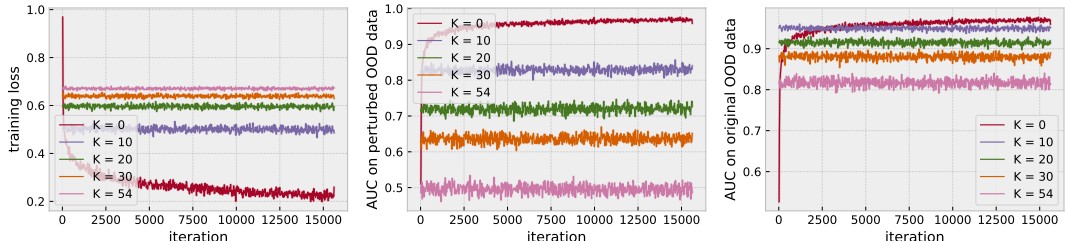

Figure 7: CIFAR-10 training curves of $\lambda = 0.1$. Left: training loss curves, middle: AUROC curves ($p_t$ vs. $p_k$), and right: AUC curves ($p_{-k}$ vs. $p_k$).

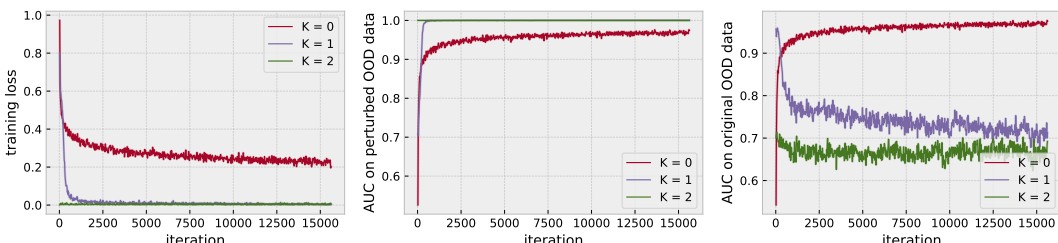

Figure 8: CIFAR-10 training curves of a failed training instance ($\lambda = 0.6$).

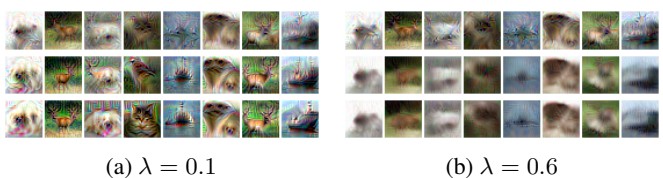

(a) $\lambda = 0.1$            (b) $\lambda = 0.6$

Figure 9: Generated samples after training with an increasing sequence of $K$ values ($K = 0, 1, 2$). Sample quality improved when $\lambda = 0.1$, but didn't when $\lambda = 0.6$.

# J   EXPANDED OOD RESULTS

## J.1   CIFAR-10

Table 15: The performances of CIFAR-10 $K = 25$ model under PGD attacks of different combinations of steps and step size. The perturbation limit is $\epsilon = 2.0$ ($L^2$ norm). Each entry is computed using 500 positive samples and 500 negative samples.

OOD dataset: Gaussian noise

| steps | 2.0 | 1.0 | 0.5 | 0.1 | 0.05 |
|---|---|---|---|---|---|
| | | | step size | | |
| 200 | 0.6781 | 0.6746 | 0.6818 | 0.7085 | 0.7194 |
| 500 | 0.6772 | 0.6717 | 0.6771 | 0.6968 | 0.7060 |
| 1000 | 0.6764 | 0.6705 | 0.6757 | 0.6923 | 0.6982 |

OOD dataset: Uniform noise

| steps | 2.0 | 1.0 | 0.5 | 0.1 | 0.05 |
|---|---|---|---|---|---|
| | | | step size | | |
| 200 | 0.9684 | 0.9674 | 0.9671 | 0.9673 | 0.9678 |
| 500 | 0.9684 | 0.9673 | 0.9669 | 0.9671 | 0.9672 |
| 1000 | 0.9684 | 0.9673 | 0.9669 | 0.9670 | 0.9670 |

OOD dataset: ImageNet

| steps | 2.0 | 1.0 | 0.5 | 0.1 | 0.05 |
|---|---|---|---|---|---|
| | | | step size | | |
| 200 | 0.5206 | 0.5171 | 0.5192 | 0.5252 | 0.5323 |
| 500 | 0.5206 | 0.5167 | 0.5182 | 0.5210 | 0.5239 |
| 1000 | 0.5199 | 0.5163 | 0.5181 | 0.5203 | 0.5222 |

OOD dataset: Bedroom-128

| steps | 2.0 | 1.0 | 0.5 | 0.1 | 0.05 |
|---|---|---|---|---|---|
| | | | step size | | |
| 200 | 0.5323 | 0.5291 | 0.5305 | 0.5388 | 0.5472 |
| 500 | 0.5315 | 0.5284 | 0.5295 | 0.5346 | 0.5368 |
| 1000 | 0.5312 | 0.5277 | 0.5292 | 0.5333 | 0.5343 |

OOD dataset: SVHN

| steps | 2.0 | 1.0 | 0.5 | 0.1 | 0.05 |
|---|---|---|---|---|---|
| | | | step size | | |
| 200 | 0.5518 | 0.5493 | 0.5517 | 0.5590 | 0.5665 |
| 500 | 0.5510 | 0.5484 | 0.5499 | 0.5546 | 0.5575 |
| 1000 | 0.5507 | 0.5480 | 0.5494 | 0.5531 | 0.5549 |

OOD dataset: CelebA-HQ

| steps | 2.0 | 1.0 | 0.5 | 0.1 | 0.05 |
|---|---|---|---|---|---|
| | | | step size | | |
| 200 | 0.5661 | 0.5631 | 0.5632 | 0.5673 | 0.5708 |
| 500 | 0.5657 | 0.5625 | 0.5626 | 0.5648 | 0.5666 |
| 1000 | 0.5653 | 0.5623 | 0.5624 | 0.5642 | 0.5653 |

OOD dataset: CIFAR-100

| steps | 2.0 | 1.0 | 0.5 | 0.1 | 0.05 |
|---|---|---|---|---|---|
| | | | step size | | |
| 200 | 0.4312 | 0.4283 | 0.4290 | 0.4340 | 0.4388 |
| 500 | 0.4310 | 0.4279 | 0.4286 | 0.4308 | 0.4327 |
| 1000 | 0.4309 | 0.4274 | 0.4280 | 0.4299 | 0.4309 |

Table 16: OOD detection performances of the CIFAR-10 $K = 0$ model on individual datasets. Each entry in this table and the following two tables is computed using 3000 positive samples and 3000 negative samples. When test $\epsilon > 0$, perturbations are computed using PGD attacks of steps 200 and step size 0.5.

| $\epsilon$-test | Gaussian noise | Uniform noise | ImageNet | Bedroom | SVHN | CelebA-HQ | CIFAR-100 | mean |
|---|---|---|---|---|---|---|---|---|
| 0.0 | 1.0000 | 0.9992 | 0.9905 | 0.9908 | 0.9956 | 0.9953 | 0.8475 | 0.9741 |
| 1.0 | 0.0000 | 0.0372 | 0.0001 | 0.0000 | 0.0000 | 0.0000 | 0.0001 | 0.0053 |
| 2.0 | 0.0000 | 0.0000 | 0.0000 | 0.0000 | 0.0000 | 0.0000 | 0.0000 | 0.0000 |

Table 17: OOD detection performances of the CIFAR-10 $K = 15$ model on individual datasets

| $\epsilon$-test | Gaussian noise | Uniform noise | ImageNet | Bedroom | SVHN | CelebA-HQ | CIFAR-100 | mean |
|---|---|---|---|---|---|---|---|---|
| 0.0 | 0.9911 | 1.0000 | 0.8593 | 0.9247 | 0.9182 | 0.8731 | 0.8036 | 0.9100 |
| 1.0 | 0.9218 | 1.0000 | 0.6918 | 0.7712 | 0.7590 | 0.7264 | 0.6156 | 0.7837 |
| 2.0 | 0.4709 | 0.9988 | 0.4065 | 0.4485 | 0.4451 | 0.5098 | 0.3503 | 0.5185 |

Table 18: OOD detection performances of the CIFAR-10 $K = 25$ model on individual datasets

| $\epsilon$-test | Gaussian noise | Uniform noise | ImageNet | Bedroom | SVHN | CelebA-HQ | CIFAR-100 | mean |
|---|---|---|---|---|---|---|---|---|
| 0.0 | 0.8993 | 0.9998 | 0.8138 | 0.8725 | 0.8721 | 0.8248 | 0.7411 | 0.8605 |
| 1.0 | 0.8310 | 0.9943 | 0.6943 | 0.7470 | 0.7510 | 0.7171 | 0.6075 | 0.7632 |
| 2.0 | 0.6845 | 0.9712 | 0.5178 | 0.5414 | 0.5660 | 0.5723 | 0.4381 | 0.6131 |

## J.2 CIFAR-10 CLASS 0

Table 19: OOD detection performances of the CIFAR-10 $K = 0$ model on individual datasets. Each entry in this table and the following two tables is computed using 1000 positive samples (the test set only has 1000 samples) and 1000 negative samples. PGD attack setting follows the CIFAR-10 experiment.

| $\epsilon$-test | Gaussian noise | Uniform noise | ImageNet | Bedroom | SVHN | CelebA-HQ | CIFAR-100 | mean |
|---|---|---|---|---|---|---|---|---|
| 0.0 | 1.0000 | 1.0000 | 0.9917 | 0.9888 | 0.9903 | 0.9996 | 0.9880 | 0.9940 |
| 1.0 | 0.0046 | 0.9100 | 0.0034 | 0.0007 | 0.0006 | 0.0167 | 0.0107 | 0.1352 |
| 2.0 | 0.0000 | 0.0588 | 0.0000 | 0.0000 | 0.0000 | 0.0000 | 0.0000 | 0.0084 |

Table 20: OOD detection performances of the CIFAR-10 $K = 15$ model on individual datasets

| $\epsilon$-test | Gaussian noise | Uniform noise | ImageNet | Bedroom | SVHN | CelebA-HQ | CIFAR-100 | mean |
|---|---|---|---|---|---|---|---|---|
| 0.0 | 0.9983 | 1.0000 | 0.9779 | 0.9819 | 0.9833 | 0.9999 | 0.9687 | 0.9872 |
| 1.0 | 0.9537 | 0.9999 | 0.9054 | 0.9055 | 0.8920 | 0.9974 | 0.8769 | 0.9330 |
| 2.0 | 0.6470 | 0.9913 | 0.6183 | 0.5727 | 0.4486 | 0.9495 | 0.5763 | 0.6863 |

Table 21: OOD detection performances of the CIFAR-10 $K = 25$ model on individual datasets

| $\epsilon$-test | Gaussian noise | Uniform noise | ImageNet | Bedroom | SVHN | CelebA-HQ | CIFAR-100 | mean |
|---|---|---|---|---|---|---|---|---|
| 0.0 | 0.9761 | 1.0000 | 0.9644 | 0.9680 | 0.9698 | 0.9998 | 0.9527 | 0.9758 |
| 1.0 | 0.9071 | 0.9963 | 0.9150 | 0.9146 | 0.9073 | 0.9979 | 0.8894 | 0.9325 |
| 2.0 | 0.7770 | 0.9787 | 0.7990 | 0.7802 | 0.7275 | 0.9841 | 0.7526 | 0.8284 |

## J.3   CELEBA-HQ-128

Table 22: The performances of CelebA-HQ-128 $K = 40$ model under PGD attacks of different combinations of steps and step size. The perturbation limit is $\epsilon = 10$ ($L^2$ norm). Each entry is computed using 500 positive samples and 500 negative samples.

OOD dataset: Gaussian noise

| steps | step size | | | | |
| | 8.0 | 4.0 | 2.0 | 1.0 | 0.5 |
| --- | --- | --- | --- | --- | --- |
| 100 | 0.9980 | 0.9980 | 0.9980 | 0.9980 | 0.9980 |
| 200 | 0.9980 | 0.9980 | 0.9980 | 0.9980 | 0.9980 |
| 500 | 0.9980 | 0.9980 | 0.9980 | 0.9980 | 0.9980 |

OOD dataset: Uniform noise

| steps | step size | | | | |
| | 8.0 | 4.0 | 2.0 | 1.0 | 0.5 |
| --- | --- | --- | --- | --- | --- |
| 100 | 1.0000 | 1.0000 | 1.0000 | 1.0000 | 1.0000 |
| 200 | 1.0000 | 1.0000 | 1.0000 | 1.0000 | 1.0000 |
| 500 | 1.0000 | 1.0000 | 1.0000 | 1.0000 | 1.0000 |

OOD dataset: ImageNet

| steps | step size | | | | |
| | 8.0 | 4.0 | 2.0 | 1.0 | 0.5 |
| --- | --- | --- | --- | --- | --- |
| 100 | 0.9933 | 0.9922 | 0.9920 | 0.9923 | 0.9929 |
| 200 | 0.9931 | 0.9920 | 0.9918 | 0.9920 | 0.9923 |
| 500 | 0.9932 | 0.9917 | 0.9916 | 0.9917 | 0.9920 |

OOD dataset: Bedroom-128

| steps | step size | | | | |
| | 8.0 | 4.0 | 2.0 | 1.0 | 0.5 |
| --- | --- | --- | --- | --- | --- |
| 100 | 0.9953 | 0.9939 | 0.9939 | 0.9943 | 0.9951 |
| 200 | 0.9952 | 0.9938 | 0.9935 | 0.9938 | 0.9943 |
| 500 | 0.9950 | 0.9937 | 0.9933 | 0.9935 | 0.9938 |

OOD dataset: SVHN

| steps | step size | | | | |
| | 8.0 | 4.0 | 2.0 | 1.0 | 0.5 |
| --- | --- | --- | --- | --- | --- |
| 100 | 0.9815 | 0.9777 | 0.9774 | 0.9793 | 0.9824 |
| 200 | 0.9808 | 0.9769 | 0.9760 | 0.9772 | 0.9793 |
| 500 | 0.9805 | 0.9764 | 0.9750 | 0.9755 | 0.9767 |

OOD dataset: CIFAR-10

| steps | step size | | | | |
| | 8.0 | 4.0 | 2.0 | 1.0 | 0.5 |
| --- | --- | --- | --- | --- | --- |
| 100 | 0.9855 | 0.9827 | 0.9822 | 0.9831 | 0.9851 |
| 200 | 0.9851 | 0.9823 | 0.9816 | 0.9820 | 0.9833 |
| 500 | 0.9848 | 0.9820 | 0.9811 | 0.9813 | 0.9820 |

OOD dataset: CIFAR-100

| steps | step size | | | | |
| | 8.0 | 4.0 | 2.0 | 1.0 | 0.5 |
| --- | --- | --- | --- | --- | --- |
| 100 | 0.9799 | 0.9758 | 0.9754 | 0.9765 | 0.9788 |
| 200 | 0.9794 | 0.9753 | 0.9745 | 0.9750 | 0.9764 |
| 500 | 0.9793 | 0.9749 | 0.9738 | 0.9739 | 0.9749 |

Table 23: OOD detection performances of the CelebA-HQ-128 $K = 0$ model on individual datasets. Each entry in this table and the following two tables is computed using 3000 positive samples and 3000 negative samples. When test $\epsilon > 0$, perturbations are computed using PGD attacks of steps 200 and step size 2.0.

| $\epsilon$-test | CIFAR-10 | Gaussian noise | Uniform noise | ImageNet | Bedroom | SVHN | CIFAR-100 | mean |
| --- | --- | --- | --- | --- | --- | --- | --- | --- |
| 0.0 | 1.0000 | 1.0000 | 1.0000 | 0.9999 | 1.0000 | 1.0000 | 1.0000 | 1.0000 |
| 5.0 | 0.0000 | 0.0000 | 0.0048 | 0.0004 | 0.0000 | 0.0000 | 0.0000 | 0.0008 |
| 10.0 | 0.0000 | 0.0000 | 0.0000 | 0.0000 | 0.0000 | 0.0000 | 0.0000 | 0.0000 |

Table 24: OOD detection performances of the CelebA-HQ-128 $K = 20$ model on individual datasets

| | CIFAR-10 | Gaussian noise | Uniform noise | ImageNet | Bedroom | SVHN | CIFAR-100 | mean |
| --- | --- | --- | --- | --- | --- | --- | --- | --- |
| 0.0 | 0.9999 | 1.0000 | 1.0000 | 1.0000 | 1.0000 | 1.0000 | 0.9999 | 1.0000 |
| 5.0 | 0.9974 | 1.0000 | 1.0000 | 0.9988 | 0.9993 | 0.9965 | 0.9959 | 0.9983 |
| 10.0 | 0.9459 | 0.9903 | 0.9996 | 0.9805 | 0.9809 | 0.9190 | 0.9291 | 0.9636 |

Table 25: OOD detection performances of the CelebA-HQ-128 $K = 40$ model on individual datasets

| | CIFAR-10 | Gaussian noise | Uniform noise | ImageNet | Bedroom | SVHN | CIFAR-100 | mean |
| --- | --- | --- | --- | --- | --- | --- | --- | --- |
| 0.0 | 0.9999 | 1.0000 | 1.0000 | 0.9999 | 1.0000 | 0.9999 | 0.9998 | 0.9999 |
| 5.0 | 0.9978 | 0.9997 | 1.0000 | 0.9990 | 0.9992 | 0.9968 | 0.9961 | 0.9984 |
| 10.0 | 0.9829 | 0.9958 | 0.9999 | 0.9925 | 0.9927 | 0.9744 | 0.9744 | 0.9875 |

## J.4 BEDROOM-128

Table 26: The performances of Bedroom $K = 40$ model under PGD attacks of different combinations of steps and step size. The perturbation limit is $\epsilon = 10$ ($L^2$ norm). Each entry is computed using 500 positive samples and 500 negative samples.

**OOD dataset: Gaussian noise**

| steps | 8.0 | 4.0 | 2.0 | 1.0 | 0.5 |
|---|---|---|---|---|---|
| | | | step size | | |
| 100 | 0.8659 | 0.8626 | 0.8899 | 0.9099 | 0.9189 |
| 200 | 0.8592 | 0.8490 | 0.8599 | 0.8871 | 0.9091 |
| 500 | 0.8552 | 0.8437 | 0.8467 | 0.8549 | 0.8762 |

**OOD dataset: Uniform noise**

| steps | 8.0 | 4.0 | 2.0 | 1.0 | 0.5 |
|---|---|---|---|---|---|
| | | | step size | | |
| 100 | 0.9791 | 0.9782 | 0.9783 | 0.9789 | 0.9798 |
| 200 | 0.9790 | 0.9779 | 0.9777 | 0.9781 | 0.9789 |
| 500 | 0.9789 | 0.9778 | 0.9774 | 0.9775 | 0.9779 |

**OOD dataset: ImageNet**

| steps | 8.0 | 4.0 | 2.0 | 1.0 | 0.5 |
|---|---|---|---|---|---|
| | | | step size | | |
| 100 | 0.8342 | 0.8297 | 0.8333 | 0.8388 | 0.8464 |
| 200 | 0.8326 | 0.8269 | 0.8285 | 0.8330 | 0.8382 |
| 500 | 0.8318 | 0.8250 | 0.8257 | 0.8280 | 0.8315 |

**OOD dataset: CIFAR-10**

| steps | 8.0 | 4.0 | 2.0 | 1.0 | 0.5 |
|---|---|---|---|---|---|
| | | | step size | | |
| 100 | 0.7413 | 0.7284 | 0.7358 | 0.7480 | 0.7656 |
| 200 | 0.7367 | 0.7222 | 0.7267 | 0.7356 | 0.7470 |
| 500 | 0.7338 | 0.7176 | 0.7205 | 0.7250 | 0.7315 |

**OOD dataset: SVHN**

| steps | 8.0 | 4.0 | 2.0 | 1.0 | 0.5 |
|---|---|---|---|---|---|
| | | | step size | | |
| 100 | 0.7856 | 0.7798 | 0.7889 | 0.8002 | 0.8162 |
| 200 | 0.7800 | 0.7719 | 0.7786 | 0.7864 | 0.7997 |
| 500 | 0.7756 | 0.7651 | 0.7700 | 0.7759 | 0.7836 |

**OOD dataset: CelebA-HQ**

| steps | 8.0 | 4.0 | 2.0 | 1.0 | 0.5 |
|---|---|---|---|---|---|
| | | | step size | | |
| 100 | 0.9033 | 0.9019 | 0.9042 | 0.9081 | 0.9134 |
| 200 | 0.9020 | 0.8996 | 0.9015 | 0.9042 | 0.9078 |
| 500 | 0.9013 | 0.8984 | 0.8993 | 0.9011 | 0.9033 |

**OOD dataset: CIFAR-100**

| steps | 8.0 | 4.0 | 2.0 | 1.0 | 0.5 |
|---|---|---|---|---|---|
| | | | step size | | |
| 100 | 0.7999 | 0.7899 | 0.7950 | 0.8062 | 0.8216 |
| 200 | 0.7954 | 0.7830 | 0.7869 | 0.7946 | 0.8050 |
| 500 | 0.7918 | 0.7778 | 0.7804 | 0.7855 | 0.7915 |

Table 27: OOD detection performances of the Bedroom-128 $K = 0$ model on individual datasets. Each entry in this table and the following two tables is computed using 3000 positive samples and 3000 negative samples. When test $\epsilon > 0$, perturbations are computed using PGD attacks of steps 200 and step size 2.0.

| $\epsilon$-test | CIFAR-10 | Gaussian noise | Uniform noise | ImageNet | SVHN | CelebA-HQ | CIFAR-100 | mean |
|---|---|---|---|---|---|---|---|---|
| 0.0 | 1.0000 | 1.0000 | 1.0000 | 0.9713 | 1.0000 | 0.9999 | 1.0000 | 0.9959 |
| 5.0 | 0.0000 | 0.0000 | 0.0010 | 0.0002 | 0.0000 | 0.0000 | 0.0000 | 0.0002 |
| 10.0 | 0.0000 | 0.0000 | 0.0000 | 0.0000 | 0.0000 | 0.0000 | 0.0000 | 0.0000 |

Table 28: OOD detection performances of the Bedroom-128 $K = 20$ model on individual datasets

| | CIFAR-10 | Gaussian noise | Uniform noise | ImageNet | SVHN | CelebA-HQ | CIFAR-100 | mean |
|---|---|---|---|---|---|---|---|---|
| 0.0 | 0.9959 | 1.0000 | 1.0000 | 0.9892 | 0.9955 | 0.9990 | 0.9946 | 0.9963 |
| 5.0 | 0.9604 | 0.9991 | 1.0000 | 0.9569 | 0.9634 | 0.9900 | 0.9623 | 0.9760 |
| 10.0 | 0.6834 | 0.9085 | 0.9984 | 0.7888 | 0.7061 | 0.8937 | 0.7251 | 0.8148 |

Table 29: OOD detection performances of the Bedroom-128 $K = 40$ model on individual datasets

| | CIFAR-10 | Gaussian noise | Uniform noise | ImageNet | SVHN | CelebA-HQ | CIFAR-100 | mean |
|---|---|---|---|---|---|---|---|---|
| 0.0 | 0.9525 | 0.9932 | 0.9993 | 0.9593 | 0.9474 | 0.9824 | 0.9578 | 0.9703 |
| 5.0 | 0.8986 | 0.9649 | 0.9967 | 0.9267 | 0.8995 | 0.9650 | 0.9114 | 0.9376 |
| 10.0 | 0.7274 | 0.8642 | 0.9749 | 0.8417 | 0.7740 | 0.9112 | 0.7651 | 0.8369 |

## K    EXPANDED GENERATION RESULTS

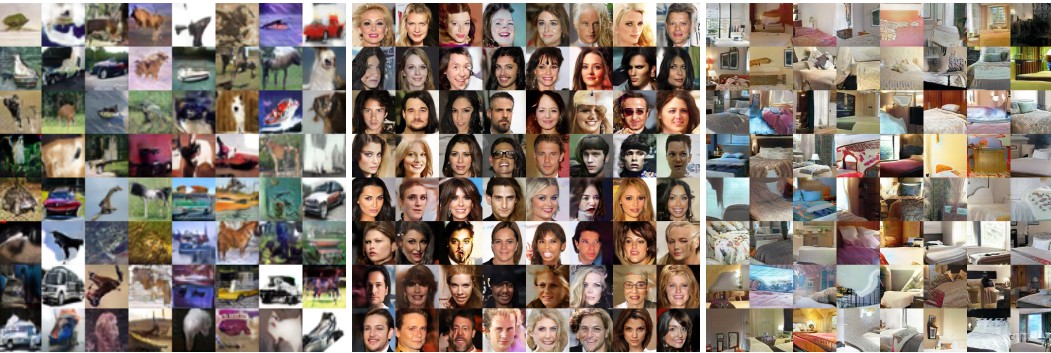

Figure 10: Samples generated by GANs (Kurach et al., 2018); results of our method are in Figure 3.

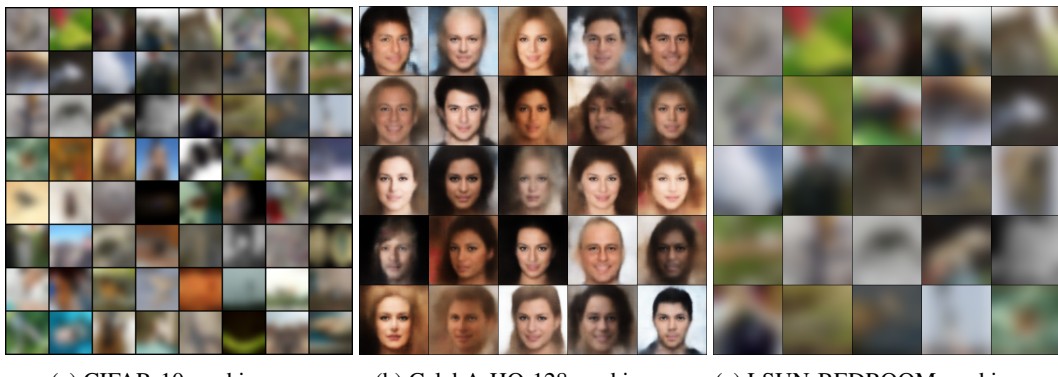

(a) CIFAR-10 seed images          (b) CelebA-HQ-128 seed images          (c) LSUN-BEDROOM seed images

Figure 11: Seed images used to generated samples in Figure 3. Seed images for CIFAR-10 and Bedroom-128 are generated by applying Gaussian blur to random images from ImageNet test set. Seed images for CelebA-HQ-128 are generated by a VAE model (Kingma, 2013; Rezende et al., 2014) trained on the CelebA-HQ-128 dataset.

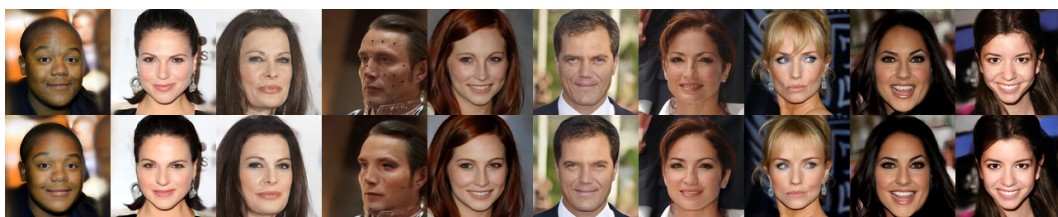

Figure 12: Face retouching results. Top row are original images from the CelebA-HQ-128 test set, and bottom row are enhanced images. The strength of retouching could be increased by performing more steps of gradient ascent on $D$.

### K.1 IMAGE GENERATION ABLATION STUDY: EFFECT OF $K$

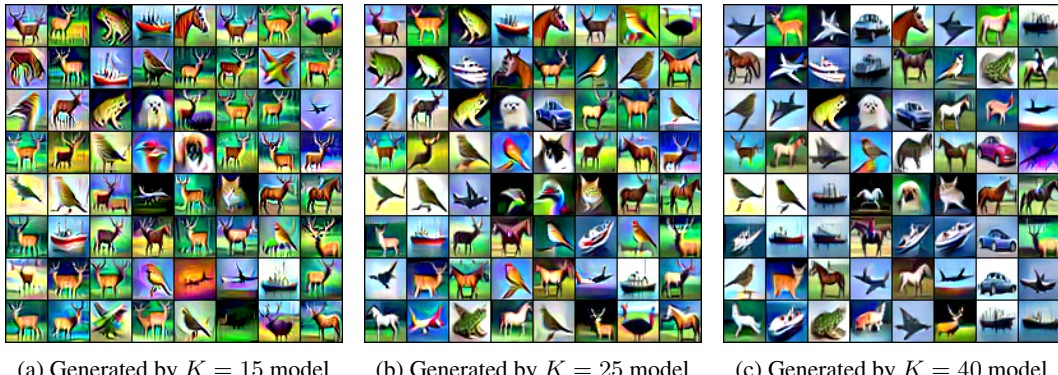

(a) Generated by $K = 15$ model     (b) Generated by $K = 25$ model     (c) Generated by $K = 40$ model

Figure 13: Samples generated by CIFAR-10 models trained with different $K$s. These generations all use the seed images in Figure 11(a).

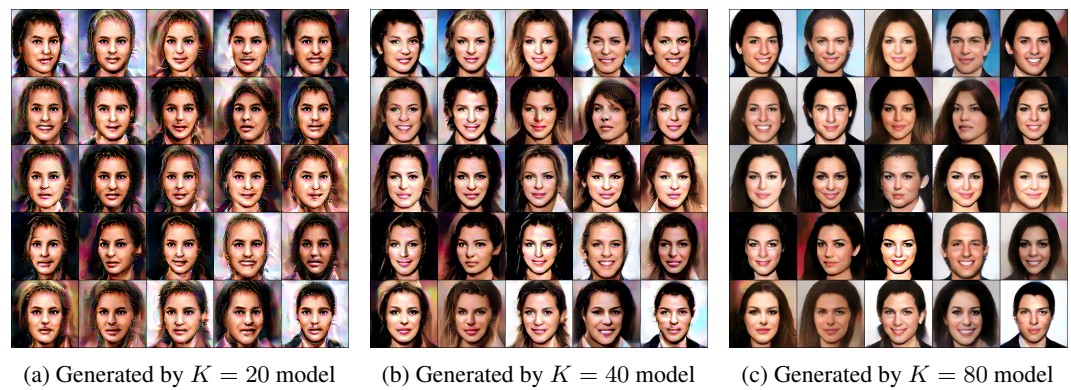

(a) Generated by $K = 20$ model     (b) Generated by $K = 40$ model     (c) Generated by $K = 80$ model

Figure 14: Samples generated by CelebA-HQ-128 models trained with different $K$s. These generations all use the seed images in Figure 11(b).

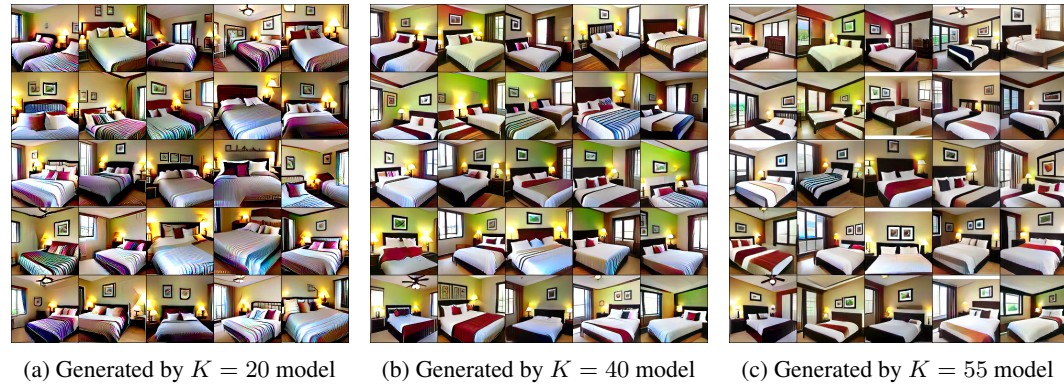

(a) Generated by $K = 20$ model     (b) Generated by $K = 40$ model     (c) Generated by $K = 55$ model

Figure 15: Samples generated by Bedroom-128 models trained with different $K$s. These generations all use the seed images in Figure 11(c).

### K.2 IMAGE GENERATION ABLATION STUDY: EFFECT OF DIFFERENT TYPES OF SEED IMAGES

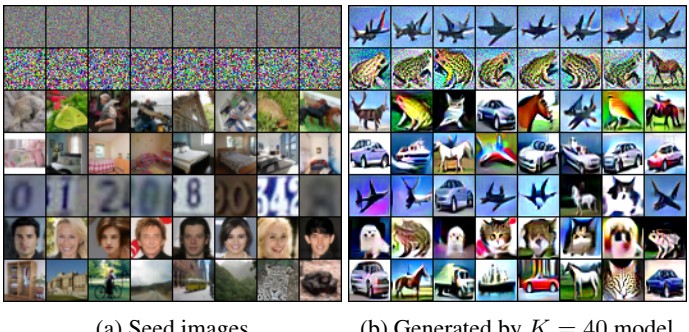

(a) Seed images  (b) Generated by $K = 40$ model

Figure 16: Samples generated by the CIFAR-10 $K = 40$ model using seed images on the left. Seed images are from OOD datasets (Table 6.

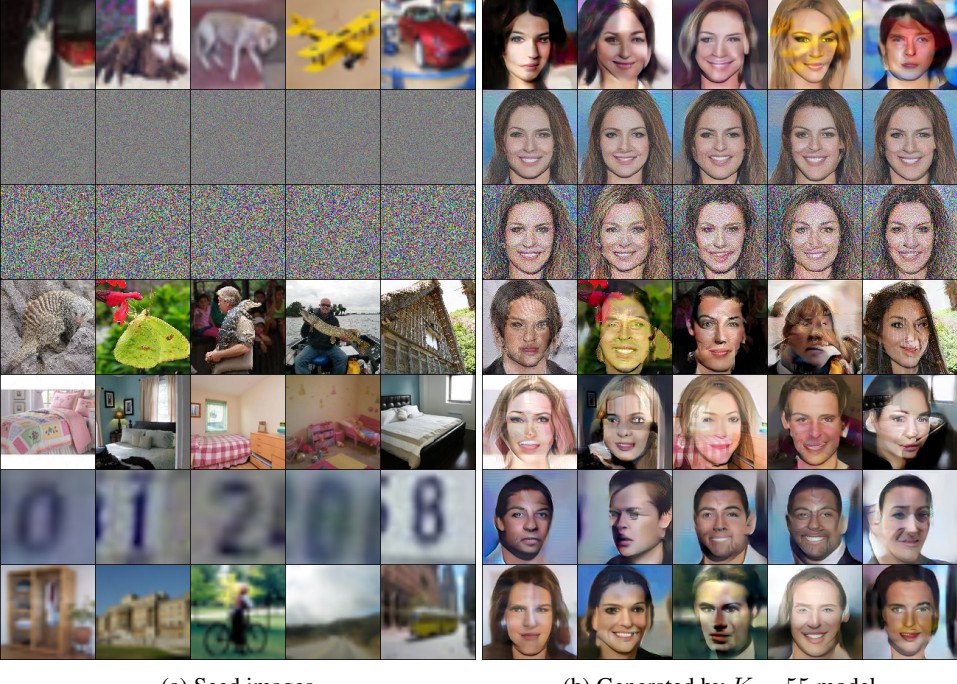

(a) Seed images  (b) Generated by $K = 55$ model

Figure 17: Samples generated by the CelebA-HQ-128 $K = 80$ model using seed images on the left. Seed images are random samples of OOD datasets (Table 6.

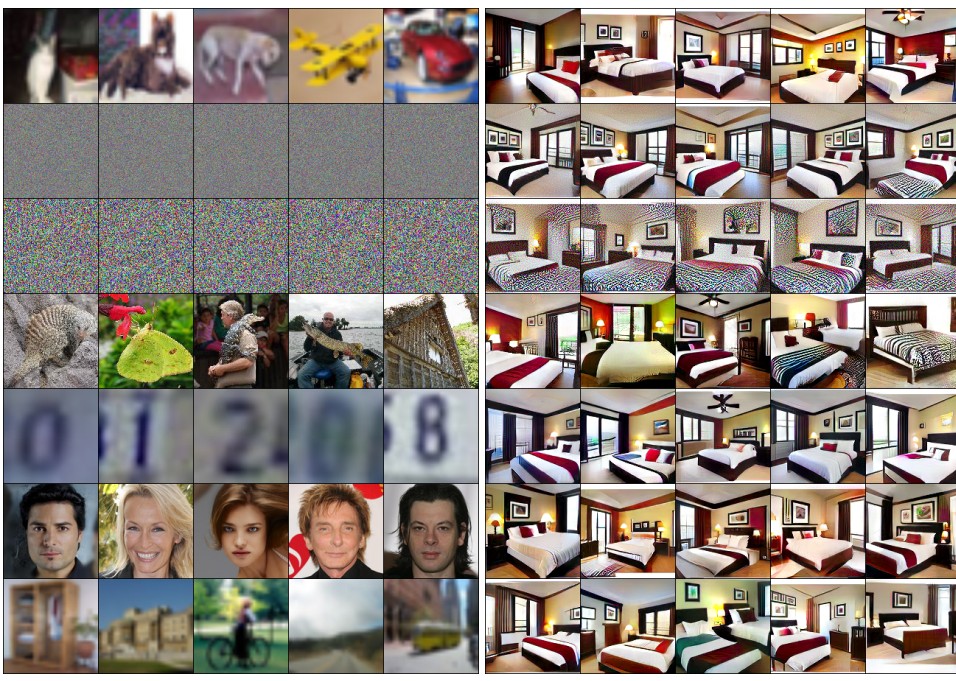

(a) Seed images     (b) Generated by $K = 55$ model

Figure 18: Samples generated by the Bedroom-128 $K = 55$ model using seed images on the left. Seed images are random samples of OOD datasets (Table 6).

## L   $256 \times 256$ RESOLUTION GENERATION

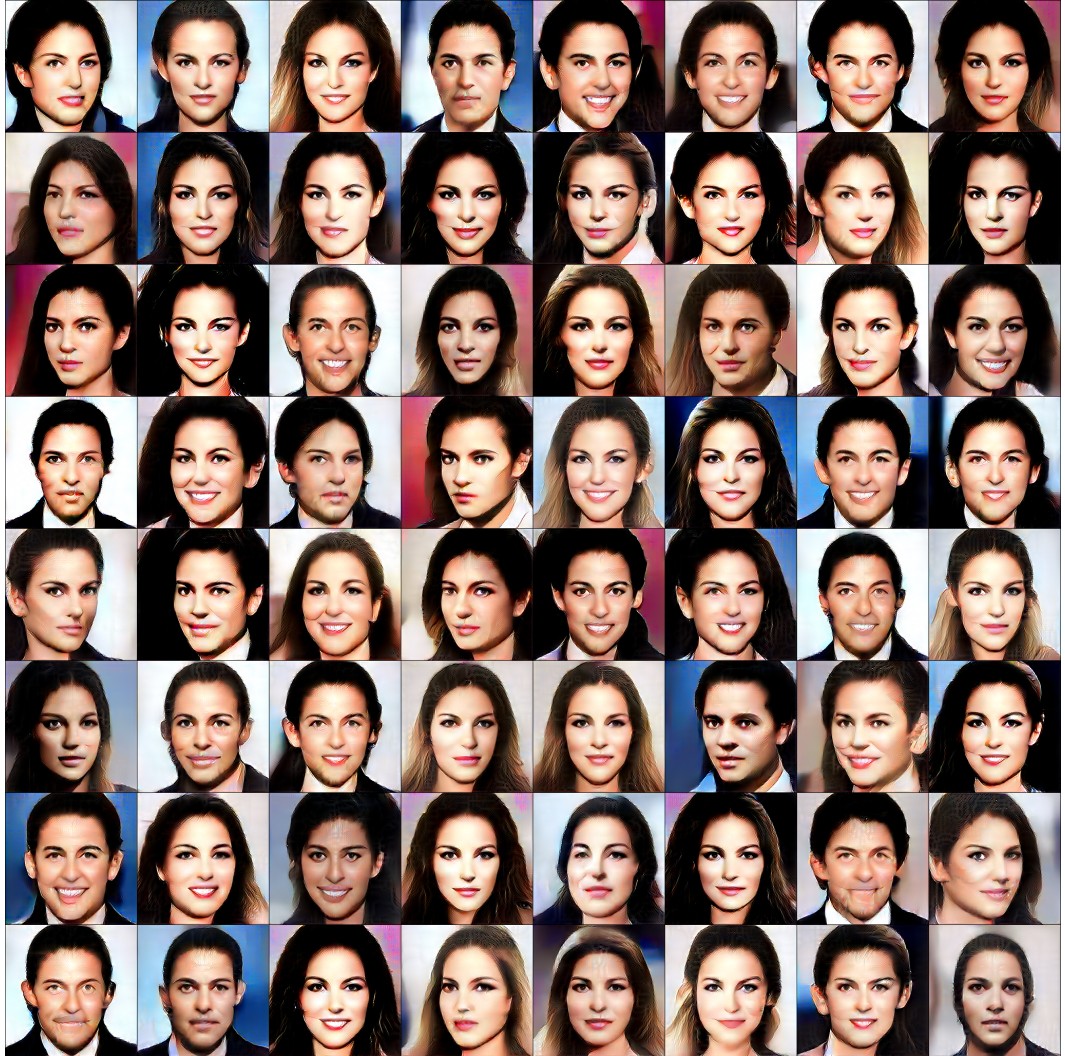

Figure 19: Uncurated $256 \times 256$ generation results in the CelebA-HQ-256 dataset.

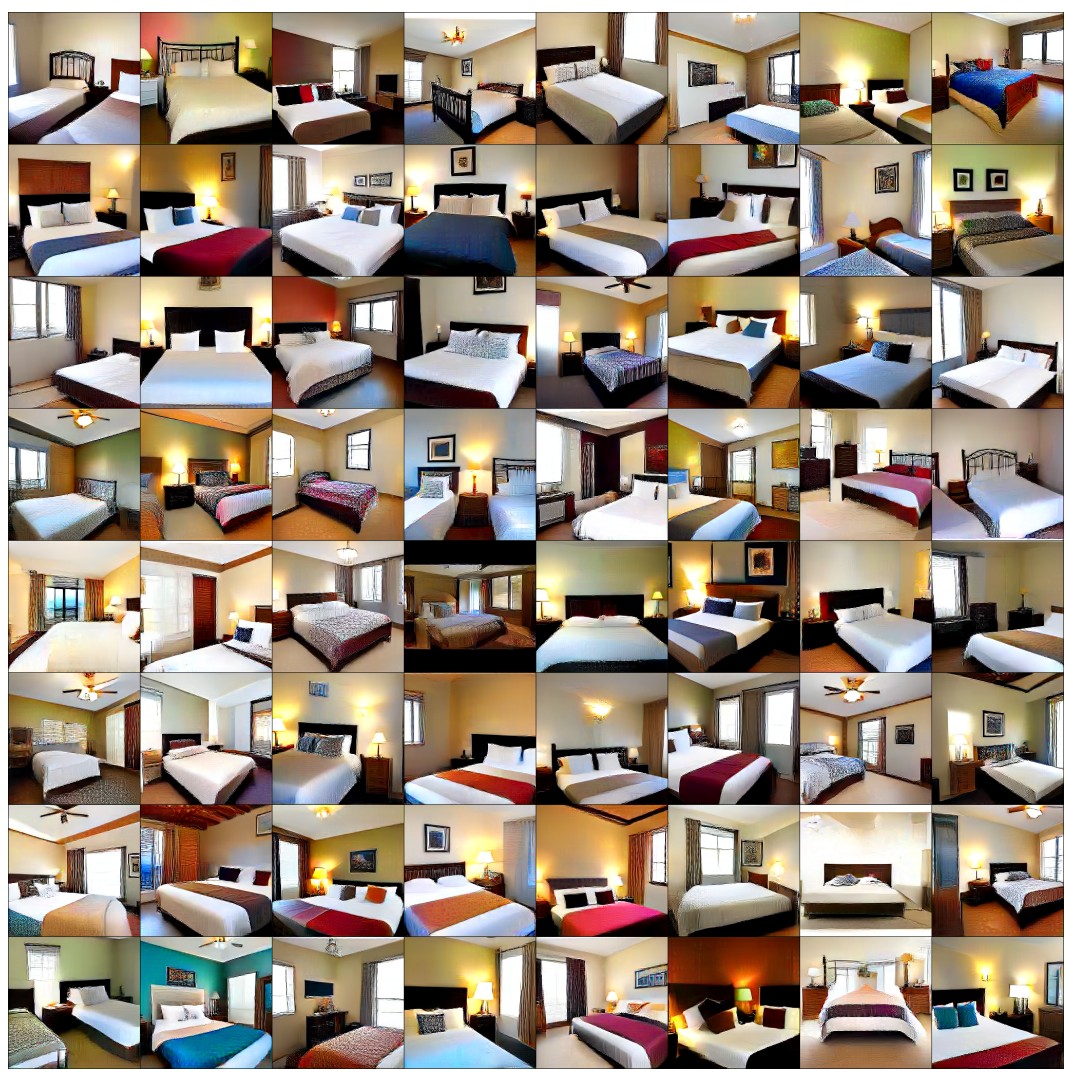

Figure 20: Uncurated $256 \times 256$ generation results in the Bedroom256 dataset. The state-of-the-art results on this dataset can be found in Figure 10 of Karras et al. (2019).

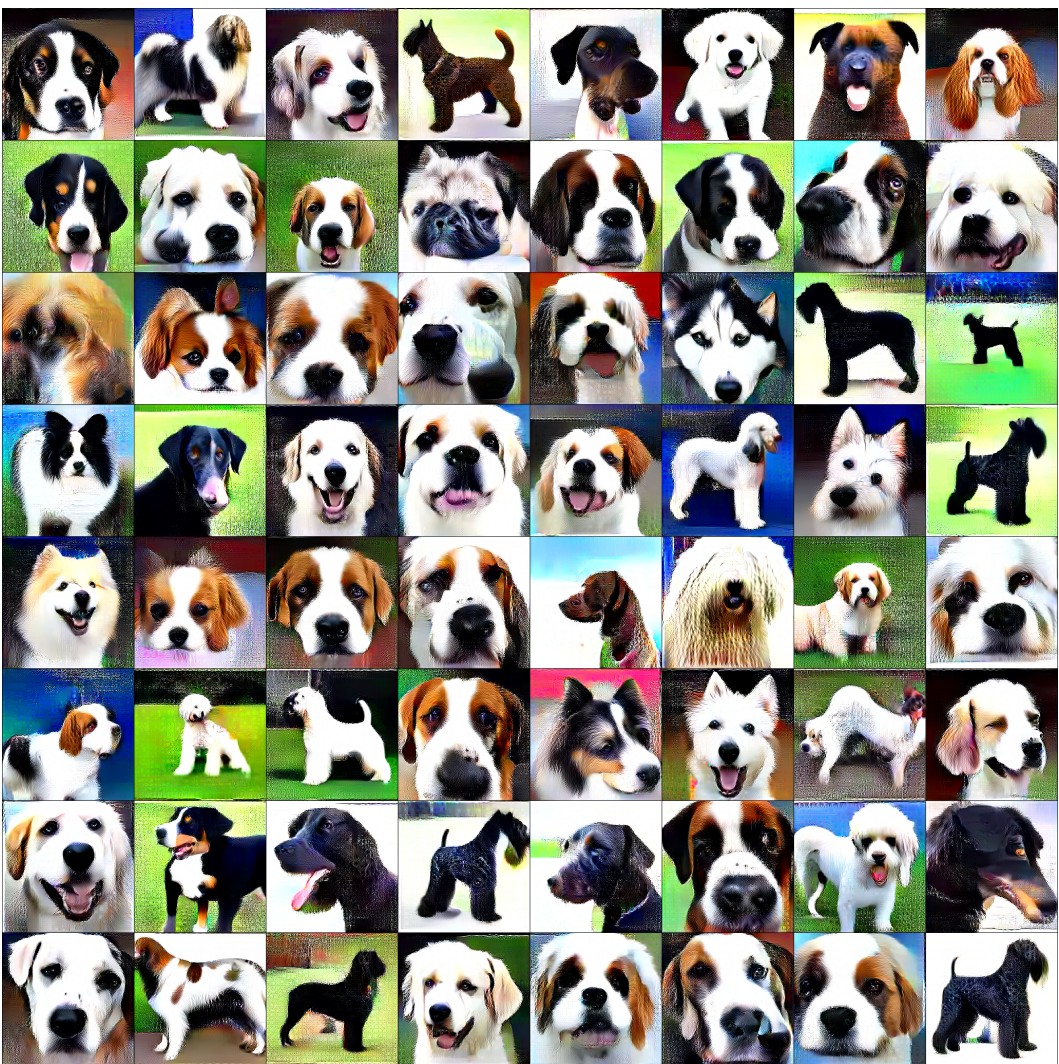

Figure 21: Uncurated $256 \times 256$ generation results in the ImageNet Dog 256 dataset. The state-of-the-art results on this dataset can be found at Brock et al. (2018), although their results are of resolution $128 \times 128$ and at the same time class-conditional. Unconditional generation results on this dataset can be found at Zhang et al. (2018).

## M   PROOF OF ALGORITHM 1'S CONVERGENCE PROPERTY

In this section we provide a proof that when $p_{-k}$ is a uniform distribution over the space $\mathcal{X} \setminus \mathrm{Supp}(p_k)$, Algorithm 1 converges to a $D$ solution with no local maxima and global maxima at support of $p_k$.

To recap Algorithm 1, in each iteration step 1 samples points from $p_k$ and $p_{-k}$, step 2 solves the inner maximization by moving samples of $p_{-k}$ to locations where $D$ has the maximum outputs, and step 3 solves the outer minimization by increasing $D$ outputs on $p_k$ samples (maximizing $\mathbb{E}_{\mathrm{x} \sim p_k}[\log D(x)]$) and decreasing outputs on $p_{-k}$ samples (maximizing $\mathbb{E}_{\mathrm{x} \sim p_t}[\log(1 - D(x))]$). Step 2 is implemented by performing gradient ascent on $D$, using initial samples of $p_{-k}$ as starting points. Given that $D$ could be a non-concave function during the course of Algorithm 1 execution, samples of $p_{-k}$ could be stuck in local maxima points in $\mathcal{X} \setminus \mathrm{Supp}(p_k)$. We now show that due to this gradient-based search method used by step 2, Algorithm 1 has the following convergence property:

**Proposition 2.** *When $p_{-k}$ is a uniform distribution over the space $\mathcal{X} \setminus Supp(p_k)$, Algorithm 1 converges to a $D$ solution with no local maxima and global maxima at support of $p_k$.*

*Proof.* We assume that $D$ has enough capacity such that step 3's update of $D$ in $\mathrm{Supp}(p_k)$ does not affect $D$'s outputs in $\mathcal{X} \setminus \mathrm{Supp}(p_k)$. We assume that the environment in which Algorithm 1 is simulated has a numeric limit of $\epsilon$ (e.g., $\epsilon = 10^{-12}$) such that $\mathcal{X} \setminus \mathrm{Supp}(p_k)$ is a finite set. (This assumption is valid when the algorithm runs on a computer.) Since $\mathcal{X} \setminus \mathrm{Supp}(p_k)$ is a finite set, we consider the case where $p_{-k}$ is a discrete uniform distribution. This distribution has non-zero probability at any point in $\mathcal{X} \setminus \mathrm{Supp}(p_k)$.

We first prove that any local maximum point in $\mathcal{X} \setminus \mathrm{Supp}(p_k)$ can be eliminated by running Algorithm 1 for a sufficient and finite number of iterations. To proceed, we first state the condition under which a local maximum point will be eliminated: a local maximum point $q$ in $\mathcal{X} \setminus \mathrm{Supp}(p_k)$ will be eliminated if via one or more iterations of the algorithm a sufficient number of $p_{-k}$ samples reach $q$. When this condition is satisfied, the cumulative effects of step 3 cause the local maximum to disappear by decreasing $D(q)$ to a sufficiently small value.

We next show that the above condition is always satisfied when Algorithm 1 runs for a finite number of iterations. Let $U$ be the set of points in $\mathcal{X} \setminus \mathrm{Supp}(p_k)$ that reach $q$ when performing gradient ascent on $D$ in step 2. $U$ is non-empty when a sufficiently small step size is used for performing the gradient ascent, as it at least contains the point $q$ itself when a step size of 0 is used. For $U$ is non-empty, a sufficient number of $p_{-k}$ samples could fall on $U$ and subsequently reach $q$ if enough samplings of $p_{-k}$ are done via step 1.

For a given $D$, the set of local maxima points in $\mathcal{X} \setminus \mathrm{Supp}(p_k)$ is a finite set. However, as new local maxima are constantly being created due to $D$'s update in each iteration, it is possible that this set will never be empty. We now prove that in a finite iterations of the algorithm this set actually goes to empty. Let $\mathcal{Q}_t$ be the set of local maxima points of $D$ in $\mathcal{X} \setminus \mathrm{Supp}(p_k)$ in iteration $t$. We have shown in the first proof that all the elements of $\mathcal{Q}_t$ are going to be reached by $p_{-k}$ samples in a finite number of iterations, hence if $\mathcal{Q}_t$ is non-empty as $t \to \infty$, $D$ values in $\mathcal{X} \setminus \mathrm{Supp}(p_k)$ will be decreased by an $\geq \epsilon$ amount for an infinite number of times, which contradicts our assumption that $D$ is a finite function in the finite set of $\mathcal{X} \setminus \mathrm{Supp}(p_k)$. □

We note that the above convergence property holds for any random initialization of $D$. However, uniform distribution is not a necessary condition here; any $p_{-k}$ distribution that has non-zero density everywhere in the data space will suffice. For a particular or particular type of initialization of $D$, its local maxima points could follow some pattern, and hence the assumption on $p_{-k}$ could be relaxed. However, in practice whether or not a $p_{-k}$ distribution is sufficient for a given $D$ can be difficult to measure.

# N    EXTENDED ADVERSARIAL OOD DETECTION RESULTS

Table 30: The performance (AUROC scores) of CIFAR-10 $K = 5$ model (the in-distribution dataset is CIFAR-10) under attacks of different configurations. Following Bitterwolf et al. (2020) we used 1000 samples for both in-distribution data and OOD data. Similarly, we used 5 random restarts to enhance the default attack, but the performance decrease is negligible.

| PGD attack steps, step size | OOD dataset (with an $L^\infty$ perturbation of $\epsilon = 0.01$) | | | |
| | Uniform Noise | Gaussian Noise | SVHN | CIFAR-100 |
|---|---|---|---|---|
| 100, 0.002 (default for Table 4) | 98.69 | 99.32 | 91.48 | 82.41 |
| 100, 0.002 (5 random restarts) | 98.69 | 99.31 | 91.45 | 82.39 |
| 500, 0.002 | 98.69 | 99.31 | 91.47 | 82.40 |
| 500, 0.005 | 98.70 | 99.32 | 91.48 | 82.43 |
| 500, 0.01 | 98.71 | 99.39 | 91.57 | 82.61 |
| 1000, 0.001 | 98.69 | 99.31 | 91.47 | 82.40 |

Table 31: Adversarial OOD detection performances (AUROC scores) when in-distribution dataset is SVHN. Performance data of methods other than ours is collected from Bitterwolf et al. (2020). The results of our method are based on an PGD attack of steps 100 and step size 0.005. For results under different attack configurations including one with random restarts see Table 32. Our SVHN model was trained with 800 Million Tiny Images dataset as the $p_{-k}$ dataset and used the ResNet18 architecture.

| Method | OOD dataset (with an $L^\infty$ perturbation of $\epsilon = 0.03$) | | | |
| | Uniform Noise | Gaussian Noise | CIFAR-10 | CIFAR-100 |
|---|---|---|---|---|
| OE (Hendrycks et al., 2018) | 98.2 | N/A | 62.5 | 60.2 |
| CCU (Meinke & Hein, 2019) | 100 | N/A | 56.8 | 52.5 |
| ACET (Hein et al., 2019) | 96.3 | N/A | 99.5 | 99.4 |
| GOOD (Bitterwolf et al., 2020) | 99.9 | N/A | 98.4 | 97.7 |
| Ours ($K = 45$) | 100 | 99.7 | 99.7 | 99.4 |

Table 32: The performance (AUROC scores) of SVHN $K = 45$ model (the in-distribution dataset is SVHN) under attacks of different configurations. Following Bitterwolf et al. (2020) we used 1000 samples for both in-distribution data and OOD data. Similarly, we used 5 random restarts to enhance the default attack, but the performance decrease is negligible.

| PGD attack steps, step size | OOD dataset (with an $L^\infty$ perturbation of $\epsilon = 0.03$) | | | |
| | Uniform Noise | Gaussian Noise | CIFAR-10 | CIFAR-100 |
|---|---|---|---|---|
| 500, 0.002 | 100 | 99.84 | 99.78 | 99.53 |
| 100, 0.005 | 100 | 99.85 | 99.78 | 99.54 |
| 100, 0.005 (5 random restarts) | 100 | 99.85 | 99.78 | 99.54 |
| 500, 0.005 | 100 | 99.85 | 99.78 | 99.54 |
| 500, 0.01 | 100 | 99.86 | 99.79 | 99.55 |
| 1000, 0.001 | 100 | 99.84 | 99.78 | 99.53 |

