# OpenReview forum: "Analyzing and Improving Generative Adversarial Training for Generative Modeling and Out-of-Distribution Detection"
_ICLR.cc/2021/Conference — Reject_

### Official Review · AnonReviewer3 · 2020-10-15
**Theoretical analysis of GAT; but need to compare with state-of-the-art and add quantitative comparison on generation.**

**Rating:** 5
**Confidence:** 5

**Review:**

The authors investigated generative adversarial training (GAT) and made two main contributions: 1) theoretically analyzed its maxmin objective and compared  with the minmax formulation used by GANs, and 2) applied GAT to OOD detection and generative models. Extensive experiments were performed for evaluating the proposed algorithm.

############################

Strong points:

. The authors analyzed the maxmin formulation of GAT theoretically by deriving the optimal solutions under different scenarios, and also showed the conditions under which the algorithm converges to the optimal solution.

. The authors pointed out the difference between the maxmin formulation and the minmax formulation used by GANs, and extended GAT for two applications: OOD detection and generative models.

. Extensive experiments were performed for evaluating the proposed algorithm.

############################

Weak points:

. In Section 3.1, for the convenience of analysis, the authors transformed equation (4) to (5). Are these two optimization problems equivalent? Seems not. Please add brief explanation.

. Algorithm 1 is claimed to solve equation (5). However, in Step 2 the maximization is still over B(x,\epsilon). It seems that Algorithm 1 is to solve equation (4). Please explain.

. Section 3.2: a practical consideration is that Step 2 cannot be perfectly solved and the authors thus proposed to use a p_{-k} that is uniformly distributed in the data space. The authors claimed that  such condition always results in no maxima and global maxima at Supp(p_k). Can this conclusion be theoretically proved? One potential limitation is that using a p_{-k} that is uniformly distributed could lead to a long training time for achieving a satisfactory performance especially in high dimensional.

. OOD detection is an active area and there are a lot of papers regarding this. In experiment, the authors mainly focused on evaluating the proposed algorithm, and did not compare it with any other OOD detection methods, e.g., log-likelihood based and likelihood ratio based. Although K = 0 can be thought of as exposing to outliers, the resulting algorithm is still under GAT framework. As OOD detection is suggested as a main application of GAT, it is important to compare it with state-of-the-art.

. It is hard to compare the quality of generation by only observing the generated images. It's better to show some quantitative comparison by using, e.g, FID score, which is commonly adopted for evaluating GANs.

####################
Some typos:

. In equation (3), should be \lambda.

---

> ### Author Response · Authors · 2020-11-17
> **Response to Reviewer3**
>
> Dear reviewer,
> Thank you very much for your time and constructive feedback! We have conducted more experiments and revised the paper to address you concerns. Below, we respond to each of your comments and look forward to your further feedback.
>
> **In Section 3.1, for the convenience of analysis, the authors transformed equation (4) to (5). Are these two optimization problems equivalent? Seems not. Please add brief explanation.**
>
> **Algorithm 1 is claimed to solve equation (5). However, in Step 2 the maximization is still over B(x,\epsilon). It seems that Algorithm 1 is to solve equation (4). Please explain.**
>
> Thanks for pointing this out. Problem (4) to (5) are equivalent when $\mathbb{B}(x,\epsilon)=\mathcal{S}$, which we have assumed (“instead of using $\epsilon$-balls imposed on individual data samples, we use the notion of a common perturbation space…” ) but didn’t explicitly stated. Similarly, we should have mentioned that Algorithm 1 solves problem (4) only when  $\mathbb{B}(x,\epsilon)=\mathcal{S}$. We have made this more explicit in the updated manuscript.
>
> **OOD detection is an active area and there are a lot of papers regarding this. In experiment, the authors mainly focused on evaluating the proposed algorithm, and did not compare it with any other OOD detection methods, e.g., log-likelihood based and likelihood ratio based. Although K = 0 can be thought of as exposing to outliers, the resulting algorithm is still under GAT framework. As OOD detection is suggested as a main application of GAT, it is important to compare it with state-of-the-art.**
>
> We completely agree. We have included a review of related work on OOD detection in appendix A. In the results section we also added a comparison with several baselines and state-of-the-art methods, for both the standard (Table 3) and adversarial (Table 4) OOD detection task.
>
> **It is hard to compare the quality of generation by only observing the generated images. It's better to show some quantitative comparison by using, e.g, FID score, which is commonly adopted for evaluating GANs.**
>
> We agree. We have included the FID scores in table 8.
>
> **Some typos:  In equation (3), should be \lambda.**
>
> Thanks, we have fixed the notation error.

---

> > ### Author Response · Authors · 2020-11-17
> > **Response to Reviewer3 (cont.)**
> >
> > **Section 3.2: a practical consideration is that Step 2 cannot be perfectly solved and the authors thus proposed to use a p_{-k} that is uniformly distributed in the data space. The authors claimed that such condition always results in no maxima and global maxima at Supp(p_k). Can this conclusion be theoretically proved? One potential limitation is that using a p_{-k} that is uniformly distributed could lead to a long training time for achieving a satisfactory performance especially in high dimensional.**
> >
> > We apologize for the confusion.  We agree that our description is misleading and we have rewritten this paragraph (Page 5).
> > As discussion in "practical considerations", step 2 cannot perfectly solve the inner problem because it is a gradient-based search method and the $D$ function could be a highly non-concave function. This issue, as our step-by-step 2D experiment demonstrates, is solved by the alternating optimizing algorithm. The introduction of uniform distribution (or any other "well distributed" data) is for the purpose of obtaining a $D$ model that has defined behavior in the whole data space, which we could not even in the 2D case when $p_{-k}$ is concentrated in a small subspace. We actually did not claim this strategy could always result in a $D$ with no local minima and global maxima at support of $p_{k}$; we meant that in several trials of the 2D experiment we consistently observed the above phenomenon.
> >
> > In addition, we did not use uniform noise to train model in our real data experiments; we have added an ablation study (Appendix H1) to demonstrate that with uniform noise as the $p_{-k}$ dataset, with the same amout of training time, we are unable to get a model that is useful for detecting real OOD data in high dimensional space.
> >
> > Regarding your question of whether we can mathematically show that when $p_{-k}$ is uniform distribution the algorithm converges to a $D$ solution with no local maxima and global maxima at support of $p_{k}$, we think it is a very good question, and it has motivated us to think deeper about this problem. After some discussion we think the answer is yes, and it is in fact quite straightforward to show that.
> >
> > ### updates
> > We have included a full poof in Appendix M of the updated manuscript.
> >
> > We thank the reviewer for raising this question; previously we only have some intuition, but now we have developed a solid theoretical understanding of this important property of the algorithm.

---

> > > ### Author Response · Authors · 2020-11-21
> > > **updates on mathematical proof**
> > >
> > > Dear Reviewer3,
> > >
> > > We have included the full proof in Appendix M of the updated manuscript (main text in Section 3.2 have also been updated accordingly).
> > >
> > > We also note that the ablation study on uniform noise in Appendix H1 seems to contradict this mathematical analysis and the 2D results. Our interpretation is that in terms of Euclidean distance, real image data live in low-dimensional manifolds that are close to each other, while uniform noise live on the unit-cube surface and is far away from real data; uniform noise is still effective for this task,  but it is far less data efficient than real image data. With uniform noise, a much larger number of inner iterations and $K$ value in Algorithm 3 may be needed to reach a satisfying detection performance. We have provided an discussion of this problem in Appendix H1.

---

### Official Review · AnonReviewer4 · 2020-10-28
**Interesting Ideas, but not ready yet.**

**Rating:** 4
**Confidence:** 4

**Review:**

**1. Summary and contributions: Briefly summarize the paper and its contributions**
This work analyzed the optimal solutions for the Generative adversarial training (GAT) and the convergence property of the training algorithm. This work also compared the minimax and maximin games, both theoretically, and empirically, with the help of a nice 2D toy example. This work also developed an unconstrained version of GAT, and evaluated it on image generation and out of distribution detection tasks.

##########################################################################

**2. Strengths: Describe the strengths of the work. Typical criteria include: soundness of the claims (theoretical grounding, empirical evaluation), significance and novelty of the contribution, and relevance to the community.**

I found the theoretical analysis to be interesting, and I especially like the 2D toy experiment in Figure 2, it strongly and clearly justified the importance of using a uniformly distributed p_(-k) distribution.

I also liked the thorough discussion of the distinction between the minimax and maximin games.

The paper has interesting ideas and a lot of content, it also introduced adversarial OOD samples, which are all very interesting to me.

##########################################################################

**3. Weaknesses: Explain the limitations of this work along the same axes as above.**

Memorization:
I think optimizing the D approach is problematic in terms of memorization, there’s nothing stopping the model to memorize the data, especially in the high dimensional space which makes the mass distribution to be very sparse.

In the Celeb A results in the middle of Figure 3, notice how the images (4,1), (4,3) look almost exactly the same, and they are also very similar to (1,3), (3,1) and (3,4).

In figure 2b and 2c, notice how the distribution all collapses to 1 point, I wonder whether this is one perspective or intuition on this problem.

Unverified claim:
At the bottom of page 7, “These results suggest that with a high capacity model and proper training, a robust OOD detection system is within reach.” where these results referred to reducing the data complexity. I think the results in this paper are not enough to make this claim. The authors’ logic here is that, if their model performs better on a simple dataset, then it implies the problem is insufficient model capacity. The assumption here is that the model can scale, which is completely unverified. Just because a model works well on MNSIT does not imply it is even possible to scale this model on ImageNet. An unverified claim like this is always a warning sign as it may mislead the readers and community.

Lack of ablation study:
I really liked the toy experiments in Figure 2, which justified the use of uniform distribution in the data space. However this only provides intuition for the higher dimensional cases, it is still necessary to conduct an ablation study to verify this is indeed the case for higher-dimensional cases, for scientific rigor.

Unfair comparison:
For image generation results, the authors’ method was compared with GANs, and the motivation is that both of the methods are trained adversarially. However, the generators of GANs never got to see the real data during train time. Here authors are optimizing D, which is trained on real data, thus I don’t think it is fair to compare with GANs.

Weak baseline for OOD (outlier exposure) and insufficient comparison:
Outlier exposure is no longer the state of the art OOD detection methods and there are so many other methods that perform better than outlier exposure on CIFAR10, for example, see Detecting Out-of-Distribution Examples with Gram Matrices: https://arxiv.org/abs/1912.12510

Lack of Related Work section:
This paper does not have a related work section. The related works are very briefly discussed in the introduction, but I think that is far from enough. I understand there is a page limit, but even putting a related work section in the appendix would be very helpful for readers to get a more complete understanding of where this work stands.

Unclear writings: See section 4.

 ##########################################################################

**4. Clarity: Is the paper well written?**
Typos:
At the top of page 3: as a results -> as a result
Bottom of page 8: a OOD -> an OOD

Ambiguity:
“elementary mass” was not defined before being used, it’d be helpful to the readers to briefly define what “elementary mass” means in this specific context.

“In order to cause more local maxima to be eliminated” could be worded better.

##########################################################################

**5. Reasons for score**
In conclusion, the ideas are very interesting but I think there is a lot more work to be done before this paper could be accepted.

---

> ### Author Response · Authors · 2020-11-17
> **Response to Reviewer4**
>
> Dear reviewer,
>
> Thank you very much for your time and constructive feedback! We have conducted more experiments and revised the paper according to your suggestions. Below, we respond to each of your comments and look forward to your further feedback.
>
> **Memorization: I think optimizing the D approach is problematic in terms of memorization, there’s nothing stopping the model to memorize the data, especially in the high dimensional space which makes the mass distribution to be very sparse.**
>
> We completely agree that our approach has the potential problem of overfitting. Our analysis says that in Figure 1 scenario 1 an ideal $D$ solution has no local maxima and global maxima at support of $p_{k}$, which means overfitting when $p_{k}$ is an empirical distribution. However, the overfitting issue is not unique to our approach. Generative models that work via the maximizing likelihood principle all have this potential problem. This is because maximizing likelihood is equivalent to minimizing the KL divergence, and a KL divergence of 0 indicates that the generated distribution and target distribution are identical – for empirical distributions, identical means the supports of the two distributions form a one-to-one match, which is exactly the definition of “memorization”. GANs is little bit different since it works by minimizing the JS divergence, but for empirical distributions a JS divergence of 0 also means that samples of two distributions form a one-to-one match. When that happens, the generator can only generate what are already in the target training data.
>
> Second, our analysis of Figure 1 scenario 1 uses two assumptions, one is that the perturbation space covers the support of $p_{k}$, and the other is that $D$ has enough capacity (which is not always true in practice especially for high dimensional complex data). Regarding the first assumption, our training algorithm (Algorithm 3) uses $K$ to control the perturbation size - this actually provides a mechanism for constraining the perturbation space and mitigating the overfitting problem. In the appendix we also have results demonstrating this effect: models trained with different $K$ tend to generate samples of different levels of fidelity.
>
> **In the Celeb A results in the middle of Figure 3, notice how the images (4,1), (4,3) look almost exactly the same, and they are also very similar to (1,3), (3,1) and (3,4).**
>
> **In figure 2b and 2c, notice how the distribution all collapses to 1 point, I wonder whether this is one perspective or intuition on this problem.**
>
> This is indeed a limitation of our approach. For the 2D case, it seems that the D model we obtained via the algorithm has a single global maximum (Figure 2b) or a few global maxima (Figure 2c). In these cases, performing gradient ascent on $D$ causes the $p_{-k}$ data to be concentrated on these maxima points. For the face experiment, it seems that the samples are stuck at local maxima points, as the resulting images do not resemble real face images; several images look the same because their seed images are trapped at the same  local maximum point. This issue tends to happen when the source images are not diverse enough. For instance, in Figure 17 we can see when source images are Gaussian  noise or uniform noise the resulting images looks quite similar.
> We think at least for the case where $D$ has no local maxima, this issue could be mitigated by properly controlling number of steps and step size when performing gradient ascent on $D$. We are also looking into the SGLD algorithm suggested by reviewer 1. We have included a discussion about this limitation on Page 9.
>
> **Unverified claim: At the bottom of page 7, “These results suggest that with a high capacity model and proper training, a robust OOD detection system is within reach.” where these results referred to reducing the data complexity. I think the results in this paper are not enough to make this claim. The authors’ logic here is that, if their model performs better on a simple dataset, then it implies the problem is insufficient model capacity. The assumption here is that the model can scale, which is completely unverified. Just because a model works well on MNSIT does not imply it is even possible to scale this model on ImageNet. An unverified claim like this is always a warning sign as it may mislead the readers and community.**
>
> We agree that the CIFAR10-classes 0 experiment is insufficient, as it only focuses on the “data complexity” side. We conducted an additional experiment where we stick to the same CIFAR10 data but used ResNet18 as the $D$ model. ResNet18 is much larger than the default model of ResNet-CIFAR in terms of disk space (43MB vs. 4.6MB), but the performance increase is only marginal (Appendix H2). Given these results we are unable to conclude that it is “simply a capacity and data complexity” issue. We have revised this paragraph and removed the unwarranted claim (Page 7).

---

> > ### Author Response · Authors · 2020-11-17
> > **Response to Reviewer4 (cont.)**
> >
> > **Lack of ablation study: I really liked the toy experiments in Figure 2, which justified the use of uniform distribution in the data space. However this only provides intuition for the higher dimensional cases, it is still necessary to conduct an ablation study to verify this is indeed the case for higher-dimensional cases, for scientific rigor.**
> >
> > We completely agree. We have conducted an ablation study (Appendix H1) where we used the CIFAR-10 class 0 data as the $p_{k}$ dataset, and respectively used ImageNet and a CIFAR-10 subset (data from class 1 – class 9) as the $p_{-k}$ dataset. ImageNet is a much larger and more diverse dataset, and it is observed that the model trained with ImageNet archives much better standard and adversarial OOD detection performance (Table 9 vs. Table 10). This confirms our intuition that in high dimensional space a large and diverse dataset should be used. We also demonstrated that (with the same amout of training time) the model didn't develop capability for detecting real image OOD inputs when uniform noise is used as $p_{-k}$ (Table 11 and Table 12).
> >
> > **Unfair comparison: For image generation results, the authors’ method was compared with GANs, and the motivation is that both of the methods are trained adversarially. However, the generators of GANs never got to see the real data during train time. Here authors are optimizing D, which is trained on real data, thus I don’t think it is fair to compare with GANs.**
> >
> > As stated in Section 5, we choose to compare with GANs because the proposed approach and GANs are closed linked (one solves the maximin problem and the other solves the minimax problem). Our evaluations are based on standard datasets, and we did not use any extra target distribution data to train our models. We also used the same model architecture and batch size to make sure setups are as close as possible. In GANs it is by design that the generator does not directly observe the data; the GANs frameworks relies on this design to learn data distribution.  Our approach, like many other generative models, requires the data to be directly observed by the module that is responsible for generation.
> > That said, we do agree that it is helpful to compare with state-of-the-art methods in the high-resolution regime. In Appendix L we have included 256x256 generation results on the CelebA-HQ, Bedroom, and ImageNet Dog dataset, and we encourage the reviewer to compare our results with results produced by state-of-the-art GANs models like StyleGAN and BigGAN.
> >
> > **Weak baseline for OOD (outlier exposure) and insufficient comparison: Outlier exposure is no longer the state of the art OOD detection methods and there are so many other methods that perform better than outlier exposure on CIFAR10, for example, see Detecting Out-of-Distribution Examples with Gram Matrices: https://arxiv.org/abs/1912.12510**
> >
> > We completely agree. We have included in the section 5.1 an comparison with more baselines and state-of-the-art methods (including the suggested one), for both the standard (Table 3) and adversarial (Table 4) OOD detection task.
> >
> > **Lack of Related Work section: This paper does not have a related work section. The related works are very briefly discussed in the introduction, but I think that is far from enough. I understand there is a page limit, but even putting a related work section in the appendix would be very helpful for readers to get a more complete understanding of where this work stands.**
> >
> > We completely agree. In appendix A we have provided a review on recent work related to out-of-distribution detection. Given the rapid development of this field, we are sure that not all related works are properly discussed. We appreciate it if the reviewer could point out missed work that we should address.
> >
> > **Unclear writings: See section 4.**
> >
> > Thanks for the feedback. We tried to improve the writing by incorporating a discussion about the suggested ablation study. We hope things are clearer now.  If the reviewer finds any other place with unclear writing, we are happy to make the changes.
> >
> > **Typos: At the top of page 3: as a results -> as a result Bottom of page 8: a OOD -> an OOD**
> >
> > Thanks, we have fixed the typo.
> >
> > **Ambiguity: “elementary mass” was not defined before being used, it’d be helpful to the readers to briefly define what “elementary mass” means in this specific context.**
> >
> > Thanks for pointing this out. “elementary mass” is a term used in the optimal transport community without rigorous definition [1]. To avoid confusion, we have replaced the term with “mass”.
> >
> > [1] Peyré, Gabriel, and Marco Cuturi. "Computational Optimal Transport: With Applications to Data Science." Foundations and Trends® in Machine Learning 11.5-6 (2019): 355-607.
> >
> > **“In order to cause more local maxima to be eliminated” could be worded better.**
> >
> > Thanks for the suggestion. We agree that this part is not clearly written. We have revised this part (page 6) by considering the suggested ablation study.

---

> > > ### Comment · AnonReviewer4 · 2020-11-25
> > > **Changing rating from 3 to 4**
> > >
> > >
> > > Thanks for the authors' reply! I have read the replies and the revised papers and decided to raise my score from 3 to 4. I understand there has been a lot of work put into this during the rebuttal period, but I still think this paper is not yet ready for the community.
> > >
> > >  I am increasing my score because of the following reasons:.
> > >
> > > **“We have revised this paragraph and removed the unwarranted claim (Page 7).”**
> > >
> > > **“We have conducted an ablation study (Appendix H1) where we used the CIFAR-10 class 0 data as the  dataset, and respectively used ImageNet and a CIFAR-10 subset (data from class 1 – class 9) as the  dataset.”**
> > >
> > > **“We have included in the section 5.1 an comparison with more baselines and state-of-the-art methods”**
> > >
> > > **“We have included in the section 5.1 an comparison with more baselines and state-of-the-art methods (including the suggested one), for both the standard (Table 3) and adversarial (Table 4) OOD detection task.”**
> > >
> > > **“In appendix A we have provided a review on recent work related to out-of-distribution detection. “**
> > >
> > > I think the related work section is pretty comprehensive, well done!
> > > Various typo fixes and clarifications.
> > >
> > > I am only increasing from 3 to 4 because of the following reasons:
> > >
> > > **Memorization**
> > >
> > > **“Generative models that work via the maximizing likelihood principle all have this potential problem. This is because maximizing likelihood is equivalent to minimizing the KL divergence, and a KL divergence of 0 indicates that the generated distribution and target distribution are identical – for empirical distributions, identical means the supports of the two distributions form a one-to-one match, which is exactly the definition of “memorization”.”**
> > >
> > > I would disagree on this point, I don’t think all generative models have this problem. If you look at the latent interpolation results from state of the art VAEs, AAEs and GANs it is clear that there is no way the model is simply memorizing the samples, as the transition between two generated samples are very smooth when interpolating in the latent space. For VAEs, the maximum likelihood objective corresponds to minimizing the Forward KL, which encourages the model’s mode covering results, which then encourage generating diverse examples, and this doesn’t really imply overfitting because the model distribution has to be smooth while the data distribution is not, so KL never goes to zero.
> > >
> > > I looked at the samples in the appendix, it seems similar problems still persist, there are many samples which look almost the same.
> > >
> > > **Unfair comparison**
> > > **“In Appendix L we have included 256x256 generation results on the CelebA-HQ, Bedroom, and ImageNet Dog dataset, and we encourage the reviewer to compare our results with results produced by state-of-the-art GANs models like StyleGAN and BigGAN.”**
> > > I’ve compared the samples, it still seems that the proposed model suffers badly from memorization as there are very similar samples. This problem should be revealed more clearly if you evaluate the samples with Fréchet Inception Distance (FID) or Inception Score (IS). This is another main reason why I think this paper is not ready for the ICLR community yet.

---

### Official Review · AnonReviewer1 · 2020-10-29

**Rating:** 7
**Confidence:** 3

**Review:**

In this paper the authors provide a theoretical and empirical analysis of the Generative Adversarial Training method (GAT) which is used to train models for OOD and adversarial example detection.

The GAT method is analyzed from a game theoretical prespective, focusing on the differences with GAN training, which are sometime conflated with it in the literature. The authors show that GAT and GAN have different training problems (maximin vs minimax) which have different optimal solutions.

The authors also propose a variant of GAT called Unconstraned GAT, which replaces the PGD attack in the inner optimization loop with an unconstrained steepest ascent update. They propose this training algorithm for both OOD detection, adversarial OOD detection and generative sampling.
They discuss the sensitivity of the proposed algorithm on the step size, which seems to critically depend on the model architecture and dataset. Experiments are performed on standard image datasets, using ImageNet as a source of known OOD examples.

Overall the work is interesting, however I find some issues:
- The key point of the theoretical anaysis comparing GAT with GAN is only made at the end. It would be better to mention in the introduction at a high level why GAT is preferable to GAN.
- Maximin and minimax are terms specific to game theory which a typical ML researcher might not be familiar with. They should be defined, possibly first in an intuitive way and then formally.
- The proposed algorithm seem quite sensitive on the step size. This might limit its practical applicability (minor issue: the step size is denoted as lambda in the text and gamma in the algorithm box).
- The proposed generative sampling procedure will likely return a mode of the distribution rather than an unbiased sample. You would need to use an SGLD-like algorithm in order get unbiased samples.

EDIT:

The revision addressed my concerns, I'm raising my evaluation to 7.

---

> ### Author Response · Authors · 2020-11-17
> **Response to Reviewer1**
>
> Dear Reviewer:
>
> Thank you very much for your time and insightful comments! We have revised the paper based on your feedback. Below we response to your questions.
>
> **The key point of the theoretical analysis comparing GAT with GAN is only made at the end. It would be better to mention in the introduction at a high level why GAT is preferable to GAN.**
>
> Thanks for the suggestion. We have included in the introduction a brief discussion about the relative merit of the proposed approach.
>
> **Maximin and minimax are terms specific to game theory which a typical ML researcher might not be familiar with. They should be defined, possibly first in an intuitive way and then formally.**
>
> Thanks for the suggestion. In Appendix B we have included a discussion about these two problems in the game theory framework. To make the materials more accessible, in Appendix C we also included a demonstration of the general strategy for solving a maximin problem.
>
> **The proposed algorithm seem quite sensitive on the step size. This might limit its practical applicability (minor issue: the step size is denoted as lambda in the text and gamma in the algorithm box).**
>
> We agree that the step size could be a potential problem. However, just as our failure mode diagnosis indicates, the problem of step size being too large can be detected in an early stage of training. We also observed that our algorithm works as long as the step size is below a certain threshold. Given these information, we think a working step size can be quickly found using binary search. Besides, in the paper we have provided the step size values for future work’s reference.
>
> Thanks, we have fixed the notation error.
>
> **The proposed generative sampling procedure will likely return a mode of the distribution rather than an unbiased sample. You would need to use an SGLD-like algorithm in order get unbiased samples.**
>
> Thanks for the suggestion. This is a known limitation of our method and we have included a discussion about this issue on Page 9. We think this issue might be mitigated by constraining the number of steps and step sizes, as the gradient-based generation process is governed by these two parameters. The suggested algorithm looks new and interesting to us, and we will definitely look into it.

---

### Author Response · Authors · 2020-11-17
**Source code**

Hi everyone,

We have submitted our source code as supplementary material. All experimental results can be reproduced using the accompanying notebooks (see README for instructions). Please let us know if you have any problem with the code.

Thanks.

---

### Author Response · Authors · 2020-11-17
**Summary of revision**

Dear reviewers,

Thank you very much for your valuable feedback! We have made our best effort to address your concerns, and we believe that the manuscript quality has improved a lot thanks to your feedback.

In the updated manuscript we have made the following changes:
- added a review of related work on out-of-distribution detection (Appendix A)
- added comparisons with more baselines and state-of-the-art methods, for both the standard (Table 3), and the adversarial OOD detection task (Table 4)
- added an ablation study on uniform noise and data diversity (Appendix H.1)
- added a discussion about the limitation of our generative modeling approach (Page 9)
- added a introductory discussion about maximin and minimax problem in game theory in Appendix B, and a demonstration of the strategy for solving a maximin problem in Appendix C
- added high-resolution (256x256) generation results on CelebA-HQ-256, Bedroom-256, and ImageNet-Dog-256 dataset (Appendix L)
- in Appendix M added a mathematical proof that when $p_{-k}$ is a uniform distribution Algorithm 1 converges to a $D$ solution with no local maxima and global maxima at support of $p_{k}$
- various writing improvements

All changes are in blue fonts.

Although the high-resolution generation results are not requested by reviewers, we think they could provide more information for evaluating our method. Our own assessment is that they are not as good as those produced by cutting edge GANs models. But we believe that these results highlight the potential of our method. In addition, they were obtained using the standard ResNet50 architecture without any architecture engineering. We note state-of-the-art Bedroom256 results can be found at StyleGAN [1] Figure 10, and ImageNet Dog 128 results can be found at BigGANs [3] (results are class-conditional, for unconditional ImageNet Dog results see [2]).

[1] Karras, Tero, Samuli Laine, and Timo Aila. "A style-based generator architecture for generative adversarial networks." Proceedings of the IEEE conference on computer vision and pattern recognition. 2019.

[2] Zhang, Han, et al. "Stackgan++: Realistic image synthesis with stacked generative adversarial networks." IEEE transactions on pattern analysis and machine intelligence 41.8 (2018): 1947-1962.

[3] Brock, Andrew, Jeff Donahue, and Karen Simonyan. "Large scale gan training for high fidelity natural image synthesis." arXiv preprint arXiv:1809.11096 (2018).

---

### Author Response · Authors · 2020-11-23
**Additional results on adversarial OOD detection**

Dear reviewers,

Our method's results in Table 4 were computed using PGD attack with particular steps and step size. This test might be inadequate as the chosen attack can hardly be the most effective one. To provide additional credentials to our method's robustness, we ran a robustness test under attacks of different PGD configurations, including one with 5 random restarts as employed by Bitterwolf et al. (2020). The test results are included in Table 30 of Appendix N. Overall we did not find any robustness issue as the results obtained with different configurations are quite similar.

In addition, we have added a comparison of our method with other methods on the SVHN dataset in Table 31, and a similar robustness test in Table 32. Consistent with the CIFAR-10 result, our method's robustness is sound and performance is competitive.

The notebooks for reproducing these results have been included in the source code.

---

### Decision · Program_Chairs · 2021-01-07
**Final Decision**

**Decision:**

Reject

**Comment:**

This paper conducts a theoretical and empirical analysis of the Generative Adversarial Training method (GAT). Although many comments have been addressed in the rebuttal, the reviewers still have few (but important) concerns, including the memorization effects and the lack of comparisons.